# UBAP2 plays a role in bone homeostasis through the regulation of osteoblastogenesis and osteoclastogenesis

Jeonghyun Kim [1,2,12], Bo-Young Kim [3,12], Jeong-Soo Lee [4,5,6,12], Yun-Mi Jeong [7,12], Hyun-Ju Cho[5], Eunkuk Park [1], Dowan Kim [1], Sung-Soo Kim [8], Bom-Taeck Kim [9], Yong Jun Choi [10], Ye-Yeon Won[11], Hyun-Seok Jin [8] ✉, Yoon-Sok Chung [10] ✉ & Seon-Yong Jeong [1,2] ✉

Osteoporosis is a condition characterized by decreased bone mineral density (BMD) and reduced bone strength, leading to an increased risk of fractures. Here, to identify novel risk variants for susceptibility to osteoporosis-related traits, an exome-wide association study is performed with 6,485 exonic single nucleotide polymorphisms (SNPs) in 2,666 women of two Korean study cohorts. The rs2781 SNP in *UBAP2* gene is suggestively associated with osteoporosis and BMD with *p*-values of $6.1 \times 10^{-7}$ (odds ratio = 1.72) and $1.1 \times 10^{-7}$ in the case-control and quantitative analyzes, respectively. Knockdown of *Ubap2* in mouse cells decreases osteoblastogenesis and increases osteoclastogenesis, and knockdown of *ubap2* in zebrafish reveals abnormal bone formation. *Ubap2* expression is associated with E-cadherin (*Cdh1*) and Fra1 (*Fosl1*) expression in the osteclastogenesis-induced monocytes. *UBAP2* mRNA levels are significantly reduced in bone marrow, but increased in peripheral blood, from women with osteoporosis compared to controls. UBAP2 protein level is correlated with the blood plasma level of the representative osteoporosis biomarker osteocalcin. These results suggest that UBAP2 has a critical role in bone homeostasis through the regulation of bone remodeling.

Osteoporosis is a multifactorial disease caused by genetic and environmental factors that requires medical intervention[1–4]. The prevalence of osteoporosis is rapidly increasing in Asian countries where the population is aging quickly[5–8]. Twin- and family-based studies suggest that 40–90% of the bone mineral density (BMD) changes are due to genetic factors[9–15]. Several bone-related biomarkers for the clinical assessment and management of osteoporosis, including diagnosis, fracture risk prediction, and/or therapy monitoring, have served as

[1]Department of Medical Genetics, Ajou University School of Medicine, Suwon, Republic of Korea. [2]Department of Biomedical Sciences, Ajou University Graduate School of Medicine, Suwon, Republic of Korea. [3]Division of Intractable Disease, Center for Biomedical Sciences, National Institute of Health, Korea Centers for Disease Control & Prevention, Cheongju, Republic of Korea. [4]Sungkyunkwan University School of Medicine, Suwon, Republic of Korea. [5]Microbiome Convergence Research Center, Korea Research Institute of Bioscience and Biotechnology, Daejeon, Republic of Korea. [6]KRIBB School, University of Science and Technology, Daejeon, Republic of Korea. [7]Disease Target Structure Research Center, Korea Research Institute of Bioscience and Biotechnology, Daejeon, Republic of Korea. [8]Department of Biomedical Laboratory Science, College of Life and Health Sciences, Hoseo University, Asan, Republic of Korea. [9]Department of Family Practice and Community Health, Ajou University School of Medicine, Suwon, Republic of Korea. [10]Department of Endocrinology and Metabolism, Ajou University School of Medicine, Suwon, Republic of Korea. [11]Department of Orthopedic Surgery, Ajou University School of Medicine, Suwon, Republic of Korea. [12]These authors contributed equally: Jeonghyun Kim, Bo-Young Kim, Jeong-Soo Lee, and Yun-Mi Jeong. ✉e-mail: jinhs@hoseo.edu; yschung@ajou.ac.kr; jeongsy@ajou.ac.kr

powerful indicators to evaluate drug treatments and to assist in the early diagnosis of osteoporosis[16,17]. Many genome-wide association studies (GWAS) have been conducted to identify novel genetic markers for osteoporosis management, resulting in the identification of numerous susceptibility loci for osteoporosis and its related traits[15,18–21]. However, these susceptibility loci have rarely been available for clinical use as biomarkers.

The advantage of GWAS is that they can identify novel susceptible single nucleotide polymorphisms (SNPs) in the whole genome. Therefore, GWAS can lead to the discovery of previously unsuspected pathological pathways, thereby providing new therapeutic targets[18,19]. To reduce the incidence of false-positives in statistical analysis-based GWAS, it is important to conduct a replication analysis in other populations and further experimental evaluation in vitro and in vivo[18]. Further, as the significance threshold of these studies is high ($p < 5 \times 10^{-8}$), SNPs with moderate significance levels tend to be unrecognized in GWAS, even if they are true functional genetic variants. Significantly associated SNPs are mainly found in intergenic or overlapping regions of genes or gene-deserted regions. This may result in difficulty in identifying the associated causative genes. The exonic SNPs located in the mature mRNA sequence, including coding regions and 5′- and 3′-untranslated regions (UTRs)[22], may be an effective strategy for identifying true susceptibility genes. Nonsynonymous SNPs lead to amino acid changes and SNPs in the 5′- and 3′-UTRs influence gene expression levels of the corresponding genes. The exonic SNPs can reduce the burden and limitations of further functional SNP analysis. Therefore, this method can be beneficial for discovering true genetic variants that are involved in protein function or the expression level of corresponding genes.

In this study, we aimed to identify novel susceptibility genes for osteoporosis and osteoporosis-related traits and then investigate their functional significance. For this purpose, our study is designed as shown in Supplementary Fig. 1. This study reports the role of UBAP2 in bone homeostasis, based on both statistical association and in vitro and in vivo functional experimental studies. We identify ubiquitin-associated protein 2 *(UBAP2)* as a novel susceptibility gene associated with osteoporosis through an exome-wide association study in women. The knockdown of *Ubap2* via RNA interference in mouse preosteoblasts and preosteoclasts decreases osteoblastogenesis and increases osteoclastogenesis, and the knockdown of *ubap2* in zebrafish with morpholino antisense oligonucleotides reveals abnormal bone formation. Human sample studies show that *UBAP2* mRNA level is significantly reduced in bone marrow, but significantly increased in peripheral blood cells, of patients with osteoporosis compared to controls. In addition, the level of UBAP2 protein is increased in blood plasma, similar to the level of osteocalcin, in patients with osteoporosis. These results suggest that UBAP2 plays a critical role in bone homeostasis through the regulation of bone remodeling.

## Results

### rs2781 in UBAP2 is suggestively associated with osteoporosis and BMD in Korean women with osteoporosis

An exome-wide association study of patients with osteoporosis and controls was conducted using PLINK version 1.07[23] and SPSS version 23.0 (IBM Corp., Armonk, NY, USA). A Bonferroni-corrected level probability ($p$) value $< 7.7 \times 10^{-6}$ was defined as a significant association. The whole-genome exonic SNPs were extracted from among all 352,228 SNPs analyzed in a previous GWAS for eight traits, including bone density, in Korean cohorts[24] using the BioMart software [http://m.ensembl.org/info/data/ biomart/][25]. An association study with 6485 exonic SNPs for osteoporosis in the Ansung cohort (discovery cohort, 312 cases, and 983 control subjects living in a rural area, Supplementary Table 1) was performed using logistic regression analysis. Seven SNPs with $p$-values < 0.001 were identified; rs2781 located in the 3′-UTR of *UBAP2* showed the lowest $p$-value ($p = 2.3 \times 10^{-4}$) (Table 1). The

seven SNPs were further analyzed in another Ansan cohort (replication cohort: 131 cases and 1240 control subjects living in an urban area) (Supplementary Table 1). rs2781 was suggestively associated with osteoporosis (OR = 1.85, $p = 9.7 \times 10^{-4}$) in the replication cohort (Table 1).

Next, we performed case-controlled association analysis in the combined subjects of both cohorts (cases, 443; controls, 2223). rs2781 was significantly associated with osteoporosis, satisfying the Bonferroni correction significance $p$-value of $6.1 \times 10^{-7}$, 95% confidence interval of 1.39–2.13, and odds ratio (OR) of 1.72 (Table 1 and Supplementary Fig. 2a). We then carried out quantitative-trait association analysis of the same exonic SNPs for BMD estimated using speed of sound (SOS) T-score at the midshaft tibia (MT) and distal radius (DR) in all women in the two cohorts, including controls and subjects with osteopenia and osteoporosis (n = 3,569; Supplementary Table 1). Only rs2781 was significantly associated with MT-SOS T-score ($p = 1.1 \times 10^{-7}$) (Supplementary Table 2 and Supplementary Fig. 2b, c).

To analyze all the available SNPs within *UBAP2*, we detected a total of 96 SNPs including 17 genotyped and 79 imputed, within the 5-kb gene boundary of *UBAP2*. The results of case-controlled and quantitative association analyzes indicated that many SNPs within *UBAP2* were suggestively associated with osteoporosis and MT BMD with variable $p$-values ($p < 0.05 - 1.1 \times 10^{-7}$), and among six exonic SNPs, two nonsynonymous and one synonymous SNPs showed suggestive associations with osteoporosis ($p < 2.4 \times 10^{-4} - 9.8 \times 10^{-5}$) and MT-SOS T-score ($p < 7.7 \times 10^{-5} - 1.3 \times 10^{-6}$) (Supplementary Table 3 and Supplementary Fig. 3). Clustered suggestive association signals in the *UBAP2* region indicate *UBAP2* as a potential susceptibility gene for osteoporosis and BMD, suggesting that it might play a crucial role in the pathogenesis of osteoporosis. We further examined whether sequence differences (C or G) in the rs2781 SNP allele located in the 3′-UTR of *UBAP2* could affect expression level changes of the luciferase report gene, but no difference was detected in our experimental system (Supplementary Fig. 4).

### *Ubap2* expression is increased during osteoblastic differentiation and reduced during osteoclastic differentiation

Next, we investigated changes in Ubap2 expression during osteoblastogenesis and osteoclastogenesis in bone remodeling. Mouse pre-osteoblast MC3T3-E1 cells and primary-cultured mouse bone marrow-derived osteoclast lineage monocytes were used for these experiments. Successfully cultured monocytes isolated from bone marrows were evaluated using specific markers (Supplementary Fig. 5). Notably, *Ubap2* mRNA expression was elevated upon inducing osteoblastic differentiation in MC3T3-E1 cells but was reduced upon inducing osteoclastic differentiation in primary monocytes in a time-dependent manner, similar to the mRNA levels of the representative genes involved in osteoblast and osteoclast differentiation (Supplementary Fig. 6). The cellular localization of endogenous Ubap2 during osteoblast and osteoclast differentiation was examined. Ubap2 was predominantly expressed in the cytosol of both osteoblasts and bone marrow-derived osteoclast lineage monocytes (Supplementary Fig. 7).

### Depletion of *Ubap2* expression causes changes in osteoblast and osteoclast differentiation in mouse cells

We next investigated whether the alteration in *Ubap2* expression can influence osteoblastic differentiation and/or osteoclastic differentiation. We generated the lentiviral-*Ubap2* short hairpin interfering RNA (shRNA) and retroviral-*Ubap2* cDNA overexpression constructs for knockdown and overexpression of *Ubap2*, respectively. Knockdown of *Ubap2* was confirmed in MC3T3-E1 cells and primary monocytes, but overexpression of *Ubap2* was only confirmed in MC3T3-E1 cells (Supplementary Fig. 8). The specificity of anti-Ubap2 antibodies that recognize different epitope regions of Ubap2 were tested using

**Table 1 | List of top exonic SNPs suggestively associated with osteoporosis as identified by exome-wide logistic regression analysis in the Korean women**

| Chr | SNP | Gene | Consequence to transcript | BP | A1 | A2 | Discovery cohort (Ansung) | | | | Replication cohort (Ansan) | | | | Combined subjects | | | |
|---|---|---|---|---|---|---|---|---|---|---|---|---|---|---|---|---|---|---|
| | | | | | | | MAF | | OR (95% CI) | Add p | MAF | | OR (95% CI) | Add p | MAF | | OR (95% CI) | Add p |
| | | | | | | | Cases (n = 312) | Controls (n = 983) | | | Cases (n = 131) | Controls (n = 1241) | | | Cases (n = 443) | Controls (n = 2224) | | |
| 9 | rs2781 | UBAP2 | 3' UTR | 3E+07 | C | G | 0.244 | 0.183 | 1.64 (1.26–2.13) | $2.3 \times 10^{-4}$ | 0.298 | 0.193 | 1.85 (1.28–2.66) | $9.7 \times 10^{-4}$ | 0.260 | 0.189 | 1.72 (1.39–2.13) | $\mathbf{6.1 \times 10^{-7}}$ |
| 2 | rs1056482 | GMCL1 | 3' UTR | 7E+07 | A | T | 0.381 | 0.314 | 1.48 (1.19–1.85) | $4.9 \times 10^{-4}$ | 0.370 | 0.343 | 0.96 (0.69–1.34) | 0.828 | 0.378 | 0.330 | 1.30 (1.09–1.57) | $4.5 \times 10^{-3}$ |
| 3 | rs6791542 | PRKAR2A | 3' UTR | 5E+07 | A | G | 0.091 | 0.050 | 2.09 (1.38–3.16) | $4.9 \times 10^{-4}$ | 0.061 | 0.065 | 0.85 (0.44–1.63) | 0.618 | 0.082 | 0.058 | 1.57 (1.12–2.21) | $9.0 \times 10^{-3}$ |
| 18 | rs607230 | LAMA1 | Nonsynonymous (K2002E) | 7E+06 | A | G | 0.205 | 0.173 | 1.62 (1.23–2.13) | $6.6 \times 10^{-4}$ | 0.187 | 0.179 | 1.01 (0.68–1.51) | 0.955 | 0.200 | 0.176 | 1.39 (1.11–1.75) | $3.8 \times 10^{-3}$ |
| 12 | rs3759300 | KIAA1551 | Synonymous (P1186P) | 3E+07 | T | C | 0.269 | 0.223 | 1.55 (1.20–1.99) | $6.6 \times 10^{-4}$ | 0.286 | 0.234 | 1.25 (0.87–1.80) | 0.224 | 0.274 | 0.229 | 1.46 (1.19–1.79) | $3.4 \times 10^{-4}$ |
| 12 | rs16941414 | CUX2 | Synonymous (I1198I) | 1E+08 | A | G | 0.099 | 0.147 | 0.56 (0.40–0.79) | $8.0 \times 10^{-4}$ | 0.115 | 0.142 | 0.83 (0.51–1.33) | 0.428 | 0.104 | 0.144 | 0.63 (0.48–0.83) | $1.2 \times 10^{-3}$ |
| 14 | rs3742842 | RGS6 | 3' UTR | 7E+07 | G | A | 0.077 | 0.053 | 2.05 (1.34–3.13) | $9.2 \times 10^{-4}$ | 0.095 | 0.068 | 1.70 (0.97–2.99) | 0.066 | 0.082 | 0.061 | 1.92 (1.37–2.69) | $1.4 \times 10^{-4}$ |

Age and residential area were included as covariates in the additive genetic model. *A1* minor allele, *A2* major allele, *Add p* additive model *p*-value, *BP* base pair, *Chr* chromosome, *CI* confidence interval, *MAF* minor allele frequency, *OR* odds ratio, *SNP* single nucleotide polymorphism, and *UTR*, untranslated region. The *p*-value below the Bonferroni-corrected significance level ($p < 7.7 \times 10^{6}$) is indicated in bold. The SNP positions are based on the NCBI Build 36 human genome assembly.

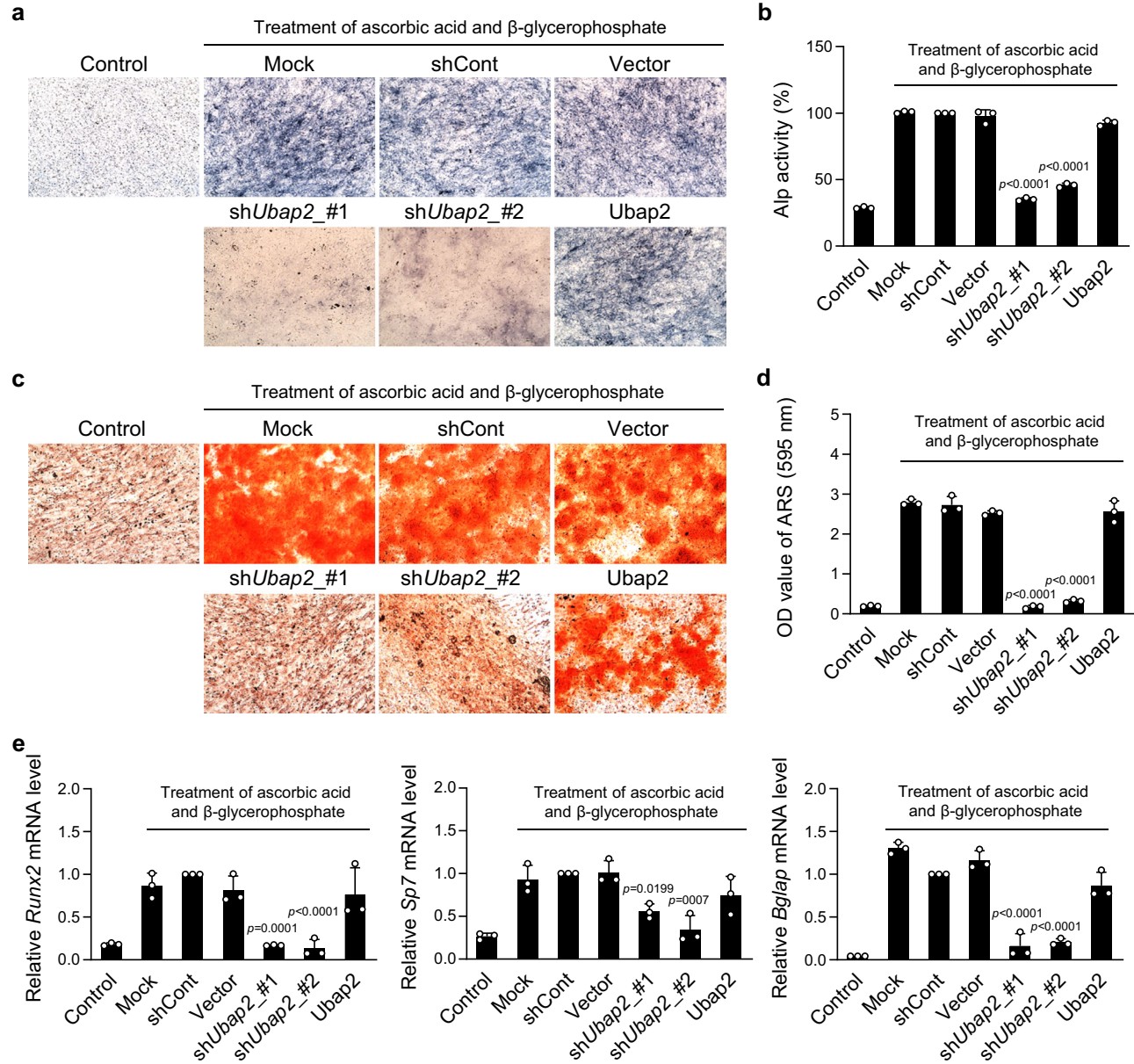

**Fig. 1 | Effects of changes in Ubap2 expression on osteoblast differentiation in pre-osteoblast MC3T3-E1 cells. a, b** Results of alkaline phosphatase (ALP) staining and ALP activity assay in *Ubap2*-knockdown and -overexpressing cells after 5 days with induction of osteoblast differentiation (*n* = 3 independent experiments; duplicate samples). **c, d** Results of mineralized nodule formation in *Ubap2*-knockdown and -overexpressing cells via Alizarin red S (ARS) staining and quantification of ARS-positive cells. Cells were incubated with the osteoblastic medium containing ascorbic acid and β-glycerophosphate for 14 days (*n* = 3 independent experiments; duplicate samples). **e** mRNA expression levels of osteoblast differentiation-related genes in *Ubap2*-knockdown and *Ubap2*-overexpressing cells. Quantitative reverse-transcription PCR was performed with the specific primers for *Runx2*, Sp7, and *Bglap*. Relative quantification of mRNA expression was presented as fold change relative to shCont (Day 5) and normalized to mouse *Gusb* expression (*n* = 3 independent experiments; duplicate samples). Control, no induction; Mock, no treatment; shCont, pLKO.1-puro empty vector; and Vector, pDON-5 Neo vector. Statistical differences of multiple groups were determined using one-way analysis of variance (ANOVA), followed by Tukey's honestly significant difference post hoc test. Exact *p*-values representing comparison to short hairpin RNA control (shCont) are shown. All data are presented as mean ± SD with individual average values. Source data are provided as a Source Data file.

western blot analysis in MC3T3-E1 cells, and their specific detection of Ubap2 was confirmed (Supplementary Fig. 9).

In pre-osteoblast MC3T3-E1 cells, *Ubap2* knockdown caused a significant reduction in the number of alkaline phosphatase (ALP)-positive cells, ALP activity, and Alizarin red S (ARS)-positive cells compared with Mock, but *Ubap2* overexpression did not affect ALP and ARS staining (Fig. 1a–d). Notably, the degree of *Ubap2* knockdown was correlated with ALP and ARS levels in cells transfected with two shRNA constructs (Fig. 1a–d and Supplementary Fig. 8). Expression levels of three osteoblastogenic markers, *Runx2*, Sp7, and *Bglap*

encoding osteocalcin (OCN), were significantly decreased in *Ubap2*-knockdown cells (Fig. 1e), demonstrating that osteoblast differentiation was suppressed by *Ubap2* depletion.

In mouse primary monocytes, *Ubap2* knockdown caused a significant increase in osteoclast differentiation. Tartrate-resistant acid-phosphatase (TRAP) staining and activity, multinuclear cell numbers, and mRNA expression of osteoclastogenic markers, such as *Acp5* and *Ctsk*, significantly increased in *Ubap2*-knockdown monocytes compared with that in Mock (Fig. 2), demonstrating that osteoclast differentiation was promoted by the *Ubap2* depletion. However, *Ubap2*

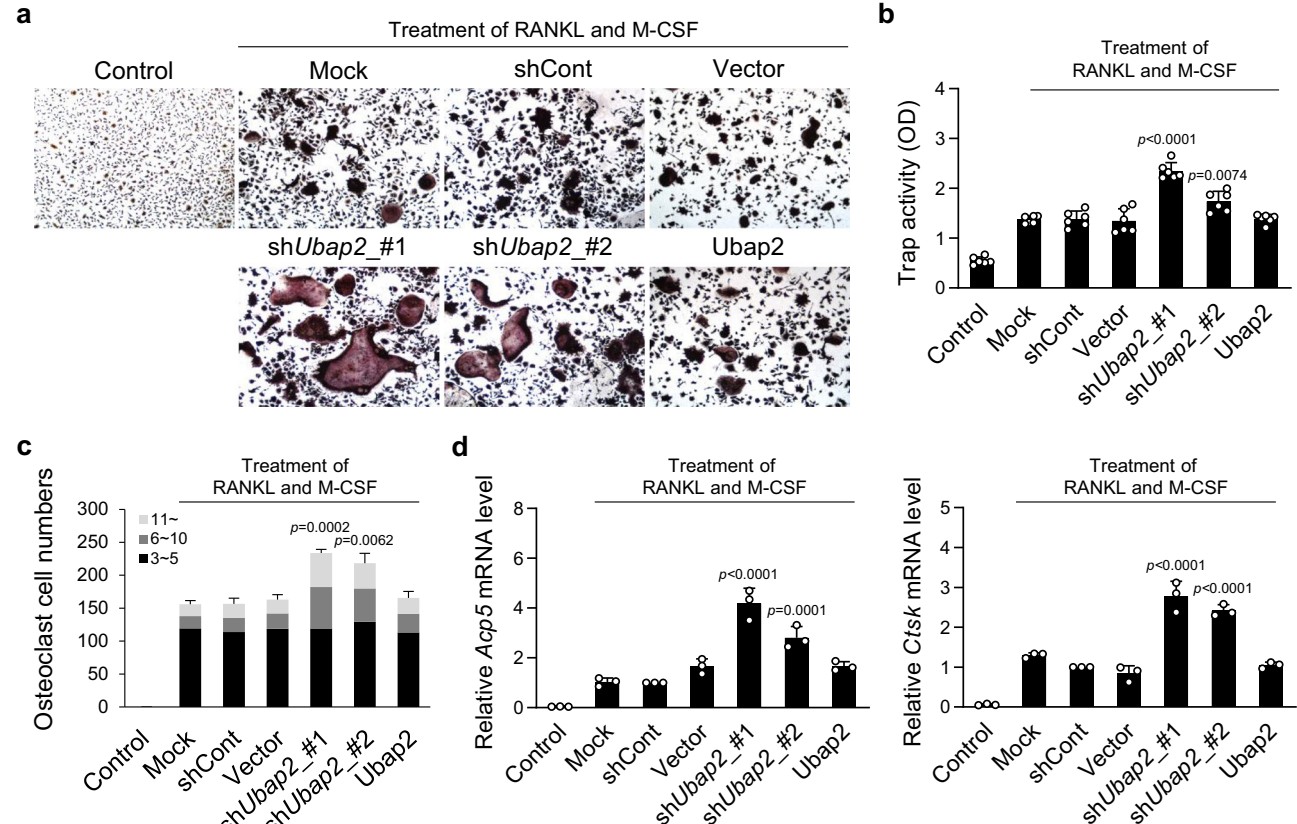

**Fig. 2 | Effects of changes in Ubap2 expression on osteoclast differentiation in primary-cultured monocytes. a, b** Results of tartrate-resistant acid-phosphatase (TRAP) staining and TRAP activity assay in *Ubap2*-knockdown and -overexpressing monocytes. Monocytes from mouse bone marrow were incubated for 4 days after induction of osteoclast differentiation (*n* = 3 independent experiments; duplicate samples). **c** The distribution of multinuclear cells. Cells were categorized according to the number of nuclei; 3 - 5, 6 - 10, and more than 11. Multinuclear cells were counted and plotted. **d** mRNA expression levels of osteoclast differentiation-related genes in *Ubap2*-knockdown and -overexpressing cells. Quantitative reverse-transcription PCR was performed with the specific primers for *Acp5* and *Ctsk*. Relative quantification of mRNA expression was presented as fold change relative to shCont (Day 4) and normalized to mouse *Gusb* expression (*n* = 3 independent experiments; duplicate samples). Control, no induction; Mock, no treatment; shCont, pLKO.1-puro empty vector; and Vector, pDON-5 Neo vector. Statistical differences of multiple groups were determined using one-way ANOVA, followed by Tukey's honestly significant difference post hoc test. Exact *p*-values representing comparison to short hairpin RNA control (shCont) are shown. All data are presented as mean ± SD. Source data are provided as a Source Data file.

overexpression did not affect osteoclast differentiation of monocytes (Fig. 2). Taken together, our experiments for *Ubap2* expression manipulation indicated that Ubap2 played crucial roles in osteoblastogenesis and osteoclastogenesis.

## *ubap2* is essential for bone and cartilage formation during the development of zebrafish

To investigate the roles of *UBAP2* in the regulation of bone remodeling in vivo, we used a zebrafish model. We identified two zebrafish homologues of human *UBAP2*, *ubap2a* and *ubap2b* by searching the Ensemble database (Ensemble gene ID: ENSDARG00000088318 for *ubap2a* and ENSDARG00000060065 for *ubap2b*). The expression patterns of *ubap2a* and *ubap2b* were examined via whole-mount in situ RNA hybridization and showed a broad expression in various regions along the developing skeletal elements. *ubap2a* mRNA was ubiquitously expressed one-day post fertilization (dpf) and predominantly expressed in the head during embryonic development, including the whole brain and the pharyngeal arches formed by cranial neural crest where the bone and cartilages begin to be generated (Supplementary Fig. 10a). The expression pattern of *ubap2b* was similar to that of *ubap2a* and was expressed in the pectoral fins, muscles, and proliferating zone of the brain (Supplementary Fig. 10b). Strong expression of both *ubap2a* and *ubap2b* mRNAs was detected in pharyngeal arches from 2 dpf and in ceratobranchials at 6 dpf (Supplementary Fig. 10a–d). The whole mount immunofluorescence analysis with anti-

Ubap2 indicated a strong expression of the Ubap2 protein in the pectoral and caudal fins that are related to bone development (Supplementary Fig. 10e, f).

Next, we conducted knockdown experiments of *ubap2a* and *ubap2b* by injecting morpholino (MO) RNAs into zebrafish embryos. Two independently designed MO RNAs for each *ubap2a* (AUG and e3i3) and *ubap2b* (e4i4 and i4e5) were examined (Supplementary Fig. 11). In the embryos injected with *ubap2a* e3i3 MO (splice-blocking), an aberrant transcript containing an intronic sequence was detected in a dose-dependent manner (Supplementary Fig. 11b, c), indicating that *ubap2a* e3i3 MO resulted in abnormal RNA splicing, leading to early termination of *ubap2a* translation. The efficacy of *ubap2a* AUG MO (translation blocking) was verified by the whole mount immunofluorescence of Ubap2 and the results revealed the reduced expression of Ubap2 protein in the zebrafish larvae at 3 dpf (Supplementary Fig. 11d–f). Similarly, injection of two independent splice-blocking MOs against *ubap2b* (e4i4 and i4e5) resulted in the generation of aberrant transcripts containing intronic sequences due to premature termination (Supplementary Fig. 11g–i).

Zebrafish endochondral bones originating from cartilage precursors develop 6 dpf and were visualized via ARS staining[26]. Two independently designed MOs for each *ubap2a* or *ubap2b* were injected into fertilized zebrafish embryos and the larvae were stained by ARS at 6 dpf. Notable, abnormal phenotypes with significantly disrupted bone formation in the pharyngeal skeleton were found in all tested *ubap2a*-

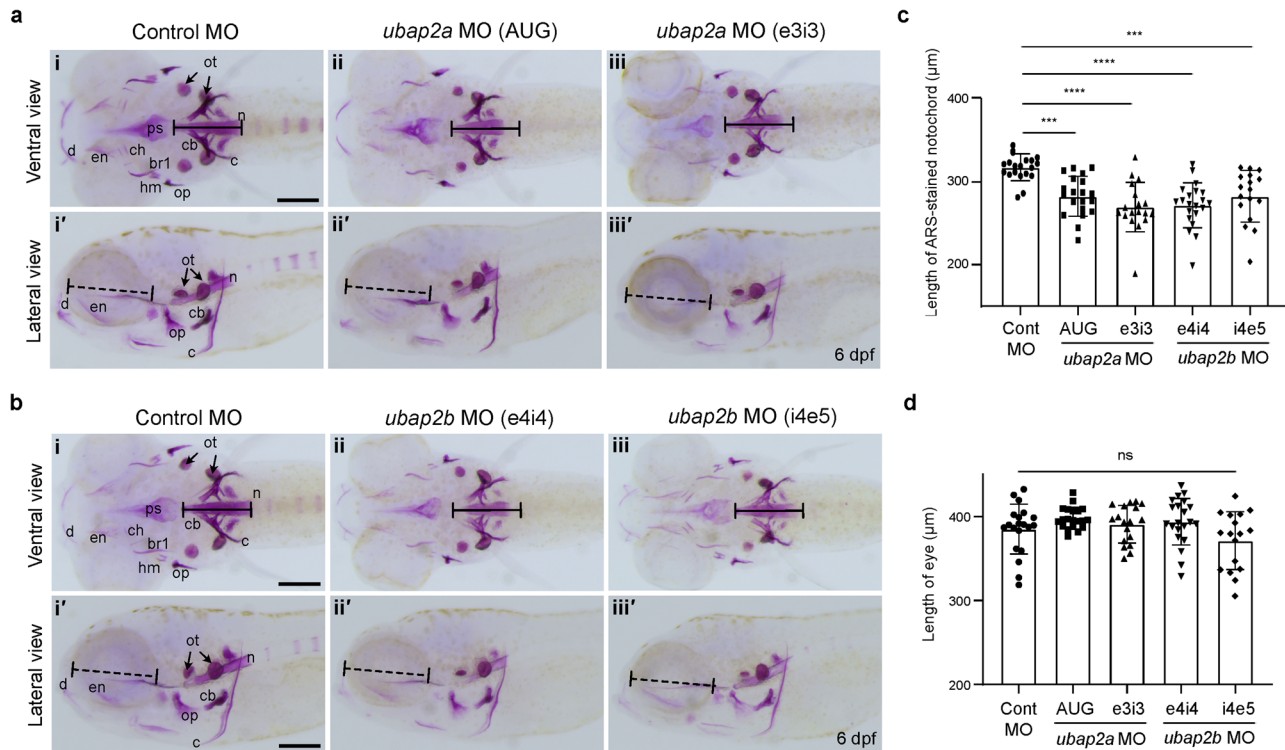

**Fig. 3 | Phenotypes of pharyngeal skeleton and notochord in the ubap2a- and ubap2b-knockdown zebrafish models during development. a** Alizarin red S (ARS) staining results of the knockdown larvae with control morpholino (Control MO, *n* = 20) (**i, i′**), *ubap2a* MO (AUG, *n* = 20) (**ii, ii′**), *ubap2a* MO (e3i3, *n* = 18) (**iii, iii′**). **b** ARS staining results of knockdown larvae with control morpholino (Control MO, *n* = 20) (**i, i′**), *ubap2b* MO (e4i4, *n* = 21) (**ii, ii′**), and *ubap2b* MO (i4e5, *n* = 17) (**iii, iii′**). Upper panels show the ventral view and lower panels show the lateral view in (**a**) and (**b**). The larvae injected with either control MO, *ubap2a*, or *ubap2b* were stained with ARS at 6 dpf and the length of the ARS-stained notochord (black lines, the lengths of the lines in each panel are identical) was measured in **a** and **b**. Eye sizes (dashed black lines) of the larvae were also measured. Concentrations of Mos; 800 µM for control MO, 500 µM for *ubap2a* AUG MO, 300 µM for *ubap2a* e3i3 MO, 800 µM for *ubap2b* e4i4 MO, and 800 µM for *ubap2b* i4e5 MO. **c, d** Quantification results of the ARS-stained notochord length (**c**) and eye diameter (**d**). Statistical significances were determined by ordinary one-way ANOVA with Tukey's test. ***$p < 0.001$ (Cont MO vs. *ubap2a* MO AUG, $p = 0.0005$; Cont MO vs. *ubap2b* MO i4e5, $p = 0.0009$ for (**c**)); ****$p < 0.0001$; ns; not significant. *n* = 17–21 biologically independent embryos per condition in (**c**) and (**d**). br1, branchiostegal ray 1; c, cleithrum; cb, ceratobranchial 5; ch, ceratohyal; d, dentary; dpf, days post fertilization; hm, hyomandibular; en, entopterygoid; m, maxilla; MO, morpholino, n, notochord; op, opercle; ot, otolith; and ps, parasphenoid. Scale bars = 200 µm. Source data are provided as a Source Data file.

and *ubap2b*-knockdown larvae, compared to the control larvae, however the calcification of otoliths in the ears was not influenced by any of the MO injections (Fig. 3a, b). Furthermore, the length of the ARS-stained notochord in *ubap2a*- and *ubap2b*-knockdown larvae was significantly shorter than the control larvae (Fig. 3c). However, the axial length of eyes was not significantly different between control and *ubap2a*- and *ubap2b*-knockdown larvae, suggesting that the shortened ARS-stained notochord's length in the *ubap2a* and *ubap2b* MO-injected larvae was not caused by the developmental delay of the whole body (Fig. 3d). Importantly, the abnormal bone phenotypes and ARS-stained notochord's length in the *ubap2a* and *ubap2b* MO-injected larvae were completely rescued by co-injection of *ubap2a* or *ubap2b* mRNA, respectively (Supplementary Fig. 12), indicating that the abnormal phenotypes in pharyngeal skeleton and notochord were specifically caused by *ubap2a* and *ubap2b* deficiencies. Taken together, the experiments in zebrafish indicate that both *ubap2a* and *ubap2b* play an essential role in the bone formation in pharyngeal skeleton during development.

During larval development between 2 and 5 dpf, zebrafish cartilages formed the craniofacial skeleton and the neurocranium, consisting of pharyngeal arches (Meckel's cartilage (jaw), ceratohyal, and five ceratobranchial arches supporting gills) and ethmoid plate[26], and cartilages were visualized by whole-mount Alcian blue staining (Supplementary Fig. 13). Treatment of fertilized zebrafish embryos with *ubap2a* e3i3 MO also significantly influenced cartilage formation in a dose-dependent manner 3.3 and 4.5 dpf, with no Alcian blue staining in

ceratobranchial arches, but the neurocranium was relatively less affected (Supplementary Fig. 13). These phenotypes indicate the essential role of *ubap2a* in the formation of cartilage as well as bone during development.

## *UBAP2* expression is significantly lower in bone marrow cells of women with osteoporosis than in controls

Next, we examined whether *UBAP2* expression is different between patients with osteoporosis and control subjects. First, we examined its expression in bone marrow samples. We collected bone marrow samples from 45 postmenopausal women who had undergone surgery due to osteoporotic fracture or osteoarthritis joint replacement [Controls (*n* = 15) and patients with osteoporosis (*n* = 30)]. The buffy coat was separated from the bone marrow and cells and used for mRNA expression quantification. The basic clinical characteristics of the recruited subjects are shown in Supplementary Table 4.

We compared the mRNA expression levels of *UBAP2* and bone remodeling-related genes encoding proteins involved in osteoblast and osteoclast differentiation in the control and the osteoporosis group. As expected, *UBAP2* mRNA expression was significantly decreased in the bone marrow of the osteoporosis group compared with that in the control group (Fig. 4a), indicating the consistent results from the above in vitro and in vivo experiments. We further tested the expression of genes associated with osteoblast or osteoclast differentiation. Expression levels of *ALPL*, *BGLAP*, and *TNF* were not significantly different between control and osteoporosis groups

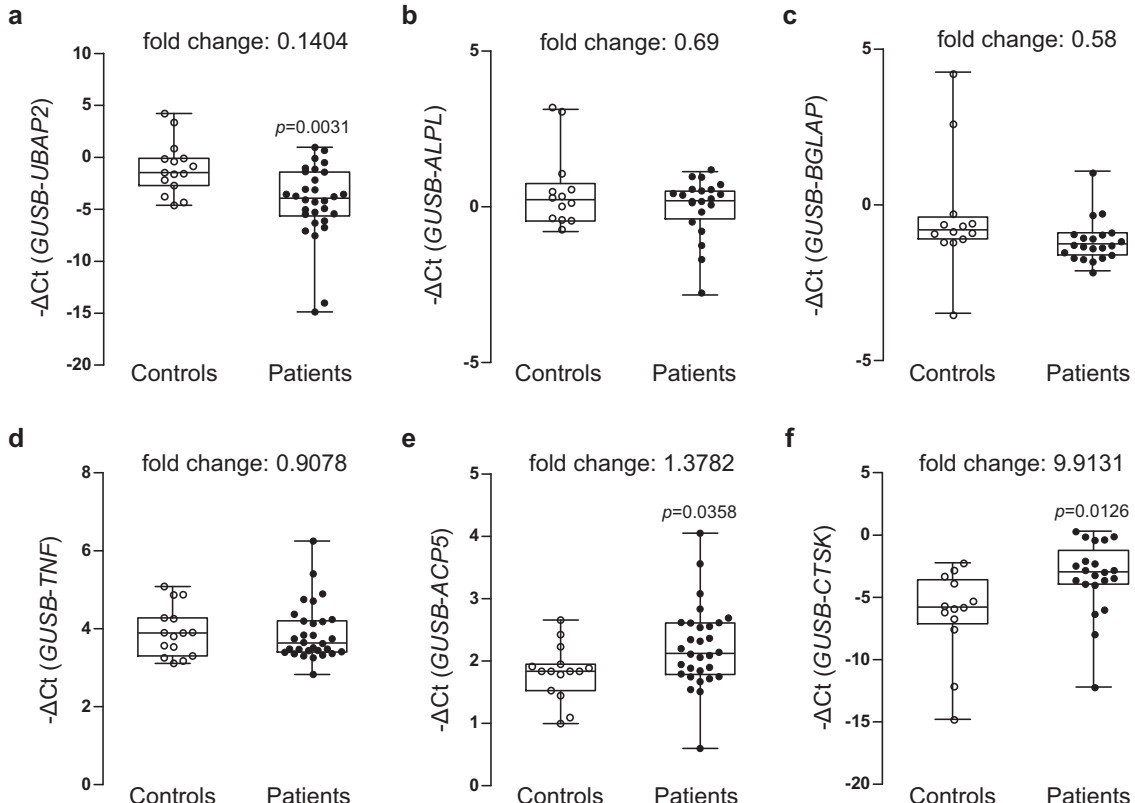

**Fig. 4 | Analysis of mRNA expression levels of UBAP2 and osteoblast and osteoclast differentiation-related genes in the bone marrow samples from postmenopausal woman controls and postmenopausal woman patients with osteoporosis.** Relative mRNA expression and fold changes in the expression of **a** *UBAP2*, osteoblast differentiation markers **b** *ALPL*, **c** *BGLAP*, **d** *TNF*, **e** *ACP5*, and **f** *CTSK* in bone marrow-derived buffy coats from controls and patients with osteoporosis are shown. Total RNAs were isolated from buffy coat of bone marrow samples from the normal postmenopausal woman controls ($n = 15$) and postmenopausal women with osteoporosis ($n = 30$) and then used as templates to synthesize cDNAs. Relative mRNA expression of the six genes relative to the internal control *GUSB* was determined via quantitative reverse-transcription PCR. The negative values of the delta cycle threshold ($-\Delta$Ct) for *UBAP2* are plotted via open and closed circles. The minimum to maximum values and median values are shown in box-and-whisker plots. All experiments were repeated three times with duplicated samples. Exact *p*-values between the groups were calculated with an unpaired two-tailed Student's *t*-test. Source data are provided as a Source Data file.

(Fig. 4b–d), but those of *ACP5* and *CTSK* were significantly increased in patients with osteoporosis compared to those in the controls (Fig. 4e, f).

## *Ubap2* expression is associated with *Cdh1* (E-cadherin) and *Fosl1* (Fra1) expression during osteoclastogenesis

Because *UBAP2* mRNA expression was correlated with the osteoclastogenesis-related genes *ACP5* and *CTSK* in the bone marrow samples of patients with osteoporosis (Fig. 4), we further investigated to identify the UBAP2 associated molecule(s) involved in osteoclast differentiation. We performed in silico pathway network analysis of related proteins with two key words 'UBAP2' and 'osteoclastogenesis' using the Ingenuity Pathway Analysis (IPA) program (QIAGEN Inc., Hilden, Germany) [https://www.qiagenbioinformatics.com/products/ingenuity-pathway-analysis]. The result of the pathway network formation showed that several molecules were directly and indirectly connected with both UBAP2 and osteoclastogenesis (Supplementary Fig. 14). Particularly, *CDH1* was directly associated with *UBAP2* and linked with several osteoclastogenesis-associated genes including *ANXA2*, *EGFR*, *TGFB1*, and *FOSL1*. Based on literature review, we selected three target genes, *ANXA2* (Annexin A2), *CDH1* (E-cadherin), and *FOSL1* (Fra1), for further studies.

Upon induction of osteoclast differentiation of mouse bone marrow-derived primary-cultured monocytes, *Cdh1* and *Fosl1* mRNA expression levels were significantly increased similar to that of the representative osteoclastogenesis-related gene *Ctsk* (Fig. 5a). Because *Anxa2* mRNA expression did not change in the osteoclastogenesis-

induced monocytes (data not shown), subsequent experiments were focused on *Cdh1* and *Fosl1*. Knockdown of *Cdh1* and *Fosl1* using shRNAs in primary-cultured monocytes was confirmed by western blotting (Supplementary Fig. 15). In the osteoclastogenesis-induced *Ubap2* knockdown monocytes, both *Cdh1* and *Fosl1* expression levels were significantly increased similar to *Ctsk* (Fig. 5b, c). Knockdown of *Cdh1* or *Fosl1* significantly reduced *Ctsk* expression resulting in the reduction of osteoclastogeneis, although *Ubap2* expression was unaltered. Notably, co-suppression of *Ubap2* and *Cdh1* or *Ubap2* and *Fosl1* by double knockdown experiments resulted in a significant reduction in *Ctsk* expression (Fig. 5b, c).

We further investigated the expression levels of *CDH1* and *FOSL1* in human samples. mRNA levels of both *CDH1* and *FOSL1* were significantly higher in bone marrow-derived buffy coat from patients with osteoporosis than that in normal controls (Fig. 5d, e). Furthermore, both E-cadherin and Fra1 proteins were significantly upregulated in patients with osteoporosis (Fig. 5f). Taken together, these results suggest that *Cdh1* and *Fosl1* are associated with *Ubap2* in the regulation of osteoclastogenesis.

## *UBAP2* mRNA and protein levels are significantly upregulated in the peripheral blood samples of women with osteoporosis

Finally, we examined UBAP2 expression levels in peripheral blood samples. To prevent the effects caused by changes in female hormone levels during the menstrual period, we recruited only 63 age-matching postmenopausal women [Controls ($n = 32$); patients with osteoporosis ($n = 31$)]. The buffy coat was separated from whole blood samples and

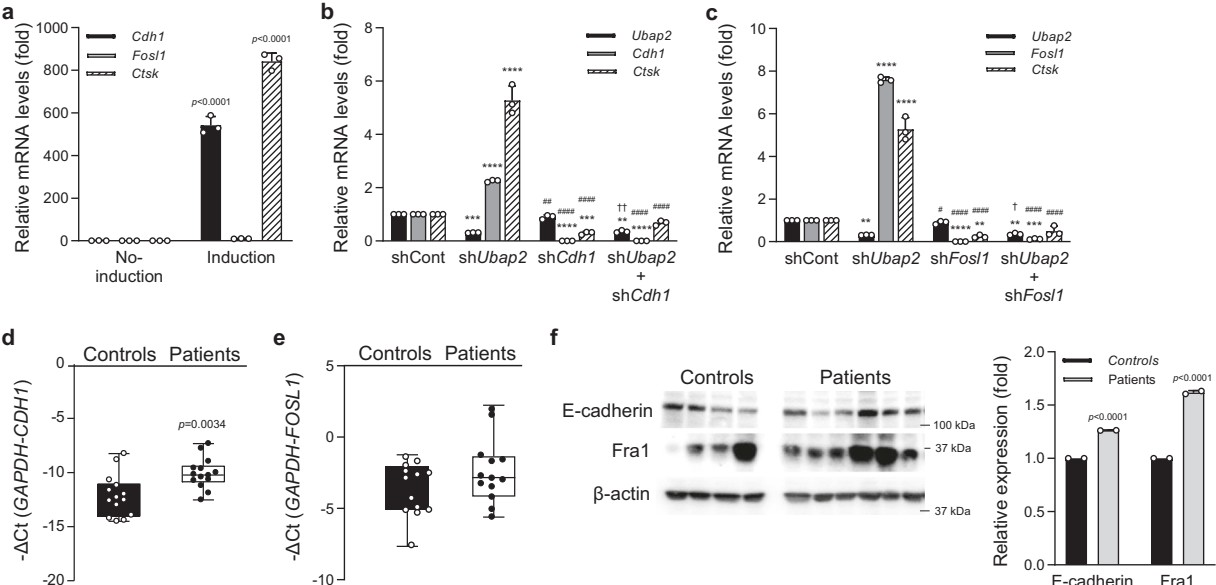

**Fig. 5 | Analysis of Cdh1 (E-cadherin) and Fosl (Fra1) expression levels in the Ubap2 knockdown cells during osteoclast differentiation and human samples of normal controls and patients with osteoporosis. a** mRNA expression levels of *Cdh1*, *Fosl1*, and *Ctsk* during osteoclast differentiation of primary cultured monocytes. Quantitative reverse-transcription PCR (qRT-PCR) was performed with the specific primers for *Cdh1*, *Fosl1*, and *Ctsk*. Relative mRNA expression was presented as fold change relative to No-induction and normalized to mouse *Gapdh* expression (*n* = 3 independent experiments; duplicate samples). Exact *p*-values of an unpaired two-tailed *t*-test are shown. **b, c** mRNA expression levels of target genes in the *Ubap2*, *Cdh1* (sh*Cdh1*_#3), and *Fosl1* (sh*Fosl1*_#3)-knockdown osteoclastogenesis-induced primary-cultured monocytes. qRT-PCR was performed as shown **a** (*n* = 3 independent experiments; duplicate samples). Statistical differences of multiple groups were determined using two-way ANOVA multiple comparisons. All data are presented as the mean ± SD. **$p$ = 0.0012 vs. shCont; ***$p$ = 0.0005 and $p$ = 0.0004

vs. shCont; ****$p$ < 0.0001 vs. shCont; ##, $p$ = 0.0035 vs. sh*Ubap2*; ####, $p$ < 0.0001 vs. sh*Ubap2*; and ††, $p$ = 0.0079 vs. sh*Cdh1* for (**b**); **, $p$ = 0.0040, $p$ = 0.0012, and $p$ = 0.0081 vs. shCont; ***$p$ = 0.0002 vs. shCont; ****$p$ < 0.0001 vs. shCont; #, $p$ = 0.0199 vs. sh*Ubap2*; ####, $p$ < 0.0001 vs. sh*Ubap2*; and †, $p$ = 0.0386 vs. sh*Fosl1* for (**c**). **d, e** mRNA expression levels of *CDH1* and *FOSL1* in bone marrow-derived buffy coats from the normal postmenopausal woman controls (*n* = 15) and post-menopausal women with osteoporosis (*n* = 15). The minimum to maximum values and median values are shown in box and whisker plots. Exact *p*-values of an unpaired two-tailed *t*-test are shown. **f** Protein expression levels of E-cadherin and Fra1 in bone marrow-derived buffy coats from controls and patients with osteoporosis. The protein levels of E-cadherin, Fra1, and β-actin were analyzed by western blotting. The intensity for quantitative analysis was normalized to β-actin using Image Processing and Analysis in Java (Image J) software [http://imagej.nih.gov/ij/]. Source data are provided as a Source Data file.

used for mRNA expression analysis. The basic characteristics of the recruited postmenopausal women are shown in Supplementary Table 5.

In contrast to the results obtained from the bone marrow samples, *UBAP2* mRNA expression in the blood samples of the postmenopausal women with osteoporosis was significantly higher than those in samples from the controls (Fig. 6a). Similar results were obtained for *ALPL* and *BGLAP* (Fig. 6b, c). Moreover, expressions of genes involved in osteoclast differentiation, *TNF*, *ACP5*, and *CTSK*, were all increased in the osteoporosis group compared to those in the normal control group (Fig. 6d–f). Finally, we examined the levels of UBAP2 and OCN proteins in the blood plasma using enzyme-linked immunosorbent assay (ELISA). Serum OCN is upregulated in patients with osteoporosis and has long been used as a diagnostic biomarker for primary osteoporosis in women[27]. The levels of UBAP2 and OCN were high in the osteoporosis group compared to those in the control group (Fig. 7), with both showing similar changes (fold changes in UBAP2 levels were correlated with those in OCN levels in the blood plasma of patients with osteoporosis). The increased level of UBAP2 in the blood plasma of patients with osteoporosis was confirmed using another ELISA kit examination (Supplementary Fig. 16).

To assess the utility of the serum UBAP2 ELISA value as a diagnostic biomarker for osteoporosis in women, receiver operating characteristic (ROC) curve regression analysis of the ELISA data of serum UBAP2 and OCN in Fig. 7 was performed using R program of the pROC package (version 1.18)[28]. The optimal cut-off points were determined as 3.033 (specificities and sensitivities: 0.524 and 0.864) for UBAP2 and 11.700 (0.857 and 0.500) for OCN, indicating that the

optimal decision threshold of serum UBAP2 and OCN between control and case groups were predicted to be 3.033 and 11.7 ng/ml (Supplementary Fig. 17), respectively. The sensitivity of UBAP2 was higher than OCN (86.4% vs. 50.0%), however the specificity was lower than OCN (52.4% vs. 85.7%). The area under the curve (AUC) values representing overall accuracy for the diagnostic tests using UBAP2 and OCN were determined as 0.727 and 0.702, respectively, indicating that the overall accuracy for osteoporosis diagnosis is comparable between the two.

## Discussion

In this study, we performed an exome-wide association study and replication analysis for osteoporosis and BMD using whole-genome exonic SNPs in two cohorts of Korean women. The previous GWAS of the SNPs, including the exonic subset of SNPs evaluated in the present study, reported that several non-coding SNPs in the *FAMC3* and *SFRP4* genes were significantly associated with the distal radius, midshaft tibia, and heel bone densities in the Korean Ansung and Ansan cohorts[24]. However, because the GWAS mainly focused on statistical *p*-values, functional SNPs and relatively less significant SNPs were ignored. At the time of study design, because we aimed to identify genetic variants with large effects on osteoporosis and/or bone-related traits, we used only exonic SNPs whose functional roles could be confirmed experimentally. Statistical analyzes of our exome-wide association study identified one suggestive SNP, rs2781, located in the 3′-UTR of *UBAP2*. The SNP showed a Bonferroni-corrected level *p*-value of $6.1 \times 10^{-7}$ and a higher OR of 1.72 (Table 1), suggesting that this genetic variant might play an important role in determining osteoporosis phenotypes. In another study, an intronic rs747091 SNP of

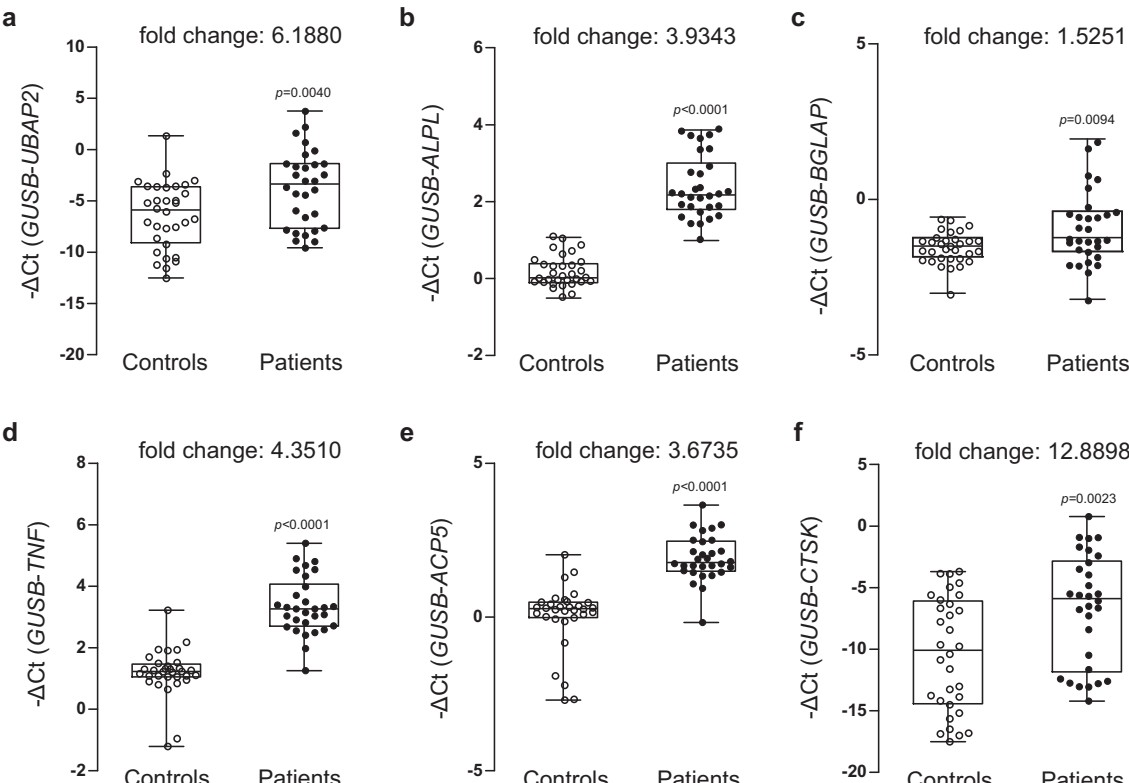

**Fig. 6 | Analysis of mRNA expression levels of UBAP2 and osteoblast and osteoclast differentiation-related genes in the peripheral blood samples from normal controls and postmenopausal women with osteoporosis.** Relative mRNA expression and fold changes in expression of **a** UBAP2, osteoblast differentiation markers **b** ALPL and **c** BGLAP, and osteoblast differentiation markers **d** TNF, **e** ACP5, and **f** CTSK between whole blood-derived buffy coats from normal controls and patients with osteoporosis are shown. Total RNAs were isolated from buffy coat of peripheral blood samples from normal control postmenopausal women (n = 32) and postmenopausal women with osteoporosis (n = 31) and then used as templates to synthesize cDNAs. Relative mRNA expression of the six genes relative to the internal control GUSB was determined by quantitative reverse-transcription PCR. The negative values of delta cycle threshold (−ΔCt) for UBAP2 are plotted by opened and closed circles. The minimum to maximum values and median values are shown in box-and-whisker plots. All experiments were repeated three times with duplicated samples. Exact p-values between the groups were calculated with an unpaired two-tailed Student's t-test. Source data are provided as a Source Data file.

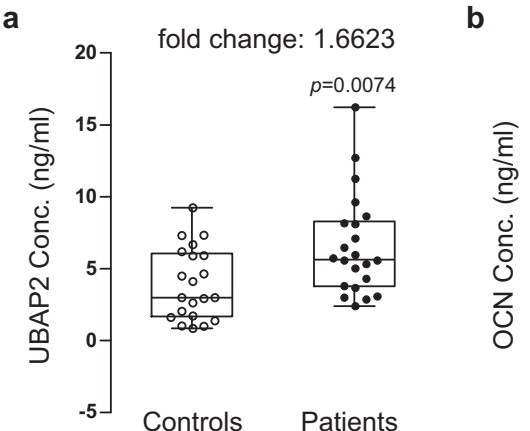

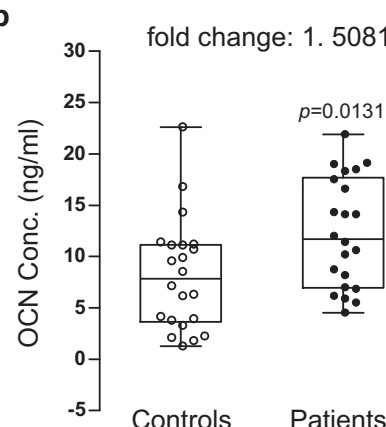

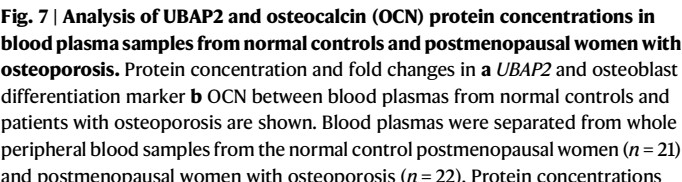

**Fig. 7 | Analysis of UBAP2 and osteocalcin (OCN) protein concentrations in blood plasma samples from normal controls and postmenopausal women with osteoporosis.** Protein concentration and fold changes in **a** UBAP2 and osteoblast differentiation marker **b** OCN between blood plasmas from normal controls and patients with osteoporosis are shown. Blood plasmas were separated from whole peripheral blood samples from the normal control postmenopausal women (n = 21) and postmenopausal women with osteoporosis (n = 22). Protein concentrations (ng/ml) in plasmas were determined by analysis of enzyme-linked immunosorbent assay data. The concentrations were plotted by opened and closed circles. The minimum to maximum concentration and median values are shown in box-and-whisker plots. Fold changes in mean values between groups are indicated. All experiments were repeated three times with duplicated samples. Exact p-values between the groups were calculated with an unpaired two-tailed Student's t-test. Source data are provided as a Source Data file.

*UBAP2* was reported to have significant association with heel BMD ($p = 9 \times 10^{-13}$ and $\beta = -0.013$)[29]. In our cohort data, this SNP also showed suggestive associations with osteoporosis ($p = 1.9 \times 10^{-4}$ and OR = 1.54) and MT-SOS T-score BMD ($p = 1.3 \times 10^{-6}$ and $\beta = -0.2$) (Supplementary Table 3). These results support the possibility that UBAP2 is a susceptible gene for osteoporosis-related traits.

Numerous genetic risk factors are responsible for causing osteoporosis and/or bone-related traits including fractures. To date, over 220 genes within approximately 200 loci have been reported based on GWAS[15]. In the GWAS catalog database [https://www.ebi.ac.uk/gwas], the intronic rs7019441 SNP of *UBAP2* showed association with a high significance level ($p = 5.0 \times 10^{-10}$) with body mass index (BMI) in another cohort. In our study cohort, rs7019441 had no relation with osteoporosis, but was significantly associated with the MT-SOS T-score: $p = 3.92 \times 10^{-4}$ ($\beta = -0.18$), indicating an inverse relation of BMI with bone density. We also performed an association study of the rs2781 SNP by adding BMI as a covariant. The significance of rs2781 for osteoporosis was $p = 2.43 \times 10^{-6}$ [OR = 1.68 (CI: 2.08-4.71)] which was slightly higher than the original results ($p = 6.1 \times 10^{-7}$) (Supplementary Table 6 and Table 1). In addition, upon adding BMI as a covariant, $p$-value ($p = 4.42 \times 10^{-7}$) of rs2781 for the quantitative trait analysis of MT-SOS T-score was similar to the original results ($p = 1.1 \times 10^{-7}$) (Supplementary Tables 3, 6), indicating that BMI may not affect osteoporosis and BMD in our study cohort.

Because most of the discovered SNPs are common variants with small effects or unelucidated biological functions, it is difficult to use these variants as biomarkers for the clinical assessment and management of osteoporosis. Therefore, many cohort studies have aimed to discover common variants with pronounced effects or less common variants with moderate effects using phenome-wide association studies and whole-genome sequencing methods[15].

To confirm the biological significance of rs2781, we performed functional analysis of *UBAP2* in vitro and in vivo. Results from functional studies revealed that lack of *Ubap2* in mouse pre-osteoblasts and mouse bone marrow-derived osteoclast lineage monocytes had deleterious effects on bone remodeling and *ubap2a* and *ubap2b* knockdowns in zebrafish caused abnormal bone formation during embryonic development (Figs. 1–3), indicating that *UBAP2* might play an essential role in bone homeostasis. However, *Ubap2* overexpression in mouse pre-osteoblasts and osteoclast-lineage monocytes did not affect osteoblast and osteoclast differentiation. This might be explained by the fact that these cells expressed a sufficient level of endogenous *Ubap2* (Supplementary Fig. 6).

We also investigated whether there was an alteration in *UBAP2* expression in samples from postmenopausal women with osteoporosis and controls. We assessed the possible expression changes of *ALPL*, *BGLAP*, *ACP5*, *CTSK*, and *TNF*, which promote osteoclast differentiation[30] using two kinds of human samples; bone marrow and peripheral blood. Bone marrow contains two lineages of stem cells; hematopoietic and mesenchymal stem cells (MSCs). Bone marrow MSCs are multipotent progenitors capable of differentiating into osteoblasts, adipocytes, chondrocytes, and possibly muscle cells[9,31]. Hence, pre-osteoblasts are derived from bone marrow MSCs. Osteoclasts are reportedly formed from peripheral blood mononuclear cells (PBMCs)[32], but another study reported that osteoclasts are formed in long-term human bone marrow cultures but not in peripheral blood cultures[33], suggesting that cells closely related to osteoclasts may be derived from PBMCs and the bone marrow.

First, we examined changes in mRNA expression in human bone marrow samples. The transcript levels of *UBAP2* were significantly downregulated, while those of *ACP5* and *CTSK* were upregulated in bone marrow samples from patients with osteoporosis compared with those in controls (Fig. 4). Unexpectedly, in blood samples, *UBAP2* expression was significantly upregulated in patients with osteoporosis (Fig. 6a). This contrasting result might be explained by previous

studies which reported that serum ALP and OCN expressions are increased in postmenopausal patients with osteoporosis compared with those in controls and that this is caused by high turnover rate during bone remodeling in patients with osteoporosis[17,27]. Our data also revealed an upregulated expression of *ALPL* and *BGLAP* in patients with osteoporosis (Fig. 6b, c). Bone resorption and formation show parallel changes[34,35]. However, bone resorption proceeds faster than bone formation during bone remodeling[36]. In other words, increased bone resorption, even when accompanied by increased bone formation, can lead to bone loss. Thus, the upregulation of UBAP2 and OCN in the blood samples from patients with osteoporosis might compensate for elevated bone resorption in postmenopausal osteoporosis (Figs. 6, 7).

Because our in vitro and in vivo data collectively indicated that the lack of UBAP2 enhanced bone resorption and inhibited bone formation, we aimed to elucidate the possible molecular mechanisms regarding the mechanism by which changes in UBAP2 expression can influence the regulation of bone remodeling. Since *UBAP2* expression changes were associated with the expression of osteoclastogenesis-related genes in human samples, we focused on the role of UBAP2 in osteoclastogenesis. In silico pathway network analysis of connected proteins with two key words 'UBAP2' and 'osteoclastogenesis' found three candidate targets, *ANXA2*, *CDH1*, and *FOSL1* (Supplementary Fig. 14).

Ubap2 reportedly forms a complex with Annexin A2 (*ANXA2*) and promotes its degradation via ubiquitination, resulting in the inhibition of hepatocellular carcinoma progression[37]. Furthermore, Annexin A2 was also reported to attenuate osteoblast growth and extracellular Annexin A2 promotes osteoclastogenesis[38]. Epithelial (E)-cadherin (*CDH1*), a type-I cadherin, is a calcium-dependent adhesion protein required for cell-cell adhesion in embryonic and adult epithelia and has important roles in epithelial cell behavior and tissue formation[39]. E-cadherin is important for cell differentiation during osteoclastogenesis[40,41]. Fra1 (*FOSL1*) a Fos-like protein and the transcriptional target of c-Fos, plays an important role in cell differentiation and tumorigenesis, and is crucial for the differentiation of several cell lineages[42]. The osteoclast differentiation factor RANKL induces transcription of *Fosl1* in a c-Fos-dependent manner and heterodimerization of Fra1 and c-Fos is essential for osteoclast differentiation[43,44].

Unexpectedly, no change in *Anxa2* expression was found by osteoclastogenic induction in the mouse bone marrow-derived primary-cultured monocytes. In contrast, upon induction of osteoclast differentiation or *Ubap2* knockdown in the primary monocytes, expression of *Cdh1* and *Fosl1* and osteoclast differentiation were significantly increased. However, knockdown of *Cdh1* or *Fosl1* resulted in the significant reduction of osteoclastogenesis (Fig. 5). Notably, co-suppression of *Ubap2* and *Cdh1* or *Ubap2* and *Fosl1* expression resulted in the significant reduction of osteoclastogenesis (Fig. 5). Since *Ubap2* knockdown caused an augmentation while *Cdh1* or *Fosl1* knockdown caused a reduction in osteoclastogenesis, these co-suppression results indicate that *Cdh1* or *Fosl1* knockdown offset the *Ubap2* knockdown effect on this process. This may be explained by the hypothesis that Ubap2 is upstream of E-cadherin and Fra1 in the osteoclastogenic signaling pathway. Because *UBAP2* encodes ubiquitin-associated protein 2 and is known to degrade Annexin A2 via ubiquitination[37], it is possible that UBAP2 is involved in the ubiquitination-dependent degradation of E-cadherin and Fra1, suggesting that UBAP2 plays a pivotal role in osteoclastogenesis through the regulation of E-cadherin and Fra1. In fact, in the bone marrow samples of patients with osteoporosis, *Cdh1* and *Fosl1* were significantly upregulated while Ubap2 was significantly downregulated (Figs. 4 and 5).

A recent review systematically summarized and discussed treatment options for osteoporosis[45]. Medications include bisphosphonates, RANKL inhibitors, estrogen agonists/antagonists, parathyroid hormone analogs, and calcitonin[45]. Bone biomarkers for therapy monitoring are

necessary for evaluating drug treatments. Serum ALP and OCN, as bone formation biomarkers, and urinary N-terminal telopeptide as a bone resorption biomarker, are commonly used[16,17]. Postmenopausal osteoporosis in women is mainly caused by excess bone resorption. The association of UBAP2 with bone formation, as well as bone resorption, is interesting. Hence, we explored whether UBAP2 might exhibit potential as a bone-formation biomarker. Serum OCN, a representative biomarker for evaluating the degree of bone formation, has been clinically used for the diagnosis of osteoporosis[27]. The Human Protein Atlas data (www. proteinatlas.org) indicate that UBAP2 is cytosolic protein and ubiquitously expressed in all tissues, including bone marrow and blood. Our study also found that Ubap2 is predominantly expressed in the cytosol of both osteoblasts and bone marrow-derived osteoclast-lineage monocytes (Supplementary Fig. 7). This result is consistent with those of two previous reports, in which UBAP2 was localized in the cytoplasm of liver cells and tumor cells and detected in both the cytoplasmic and chromatin fractions in normal human MRC5 fibroblasts[37,46]. These studies suggested the possible intracellular functions of UBAP2 as ubiquitylation regulation of RNA polymerase II as a human candidate yeast Def1 ortholog, and ubiquitination and degradation of annexin A2[37,46]. Because blood detection is a highly beneficial biomarker trait, we further tested whether UBAP2 is detectable in human blood plasma samples in patients with osteoporosis and controls. Because ALP is a biomarker for evaluating early osteoblastogenesis[17]. We used OCN as a comparable standard biomarker. ELISA-assessed protein levels in blood plasma samples revealed that both UBAP2 and OCN were significantly high in the osteoporosis group compared to the control group, and changes in the levels of UBAP2 in the two groups were closely correlated with those of OCN (Fig. 7 and Supplementary Fig. 16). Although the previously identified intracellular functions of UBAP2[37,46] are not associated with extracellular roles, our results showed that UBAP2 was detected in blood plasma by ELISA (Fig. 7 and Supplementary Fig. 16). It is unclear whether such extracellular UBAP2 results from secretion by cells or cell breakage. In comparison, how the representative osteoporosis biomarkers ALP and procollagen type 1 C-terminal propeptide (P₁CP) can be detected in serum is unknown[17].

ROC curve regression analysis is helpful for assessing the diagnostic potential of target biomarkers and for determining their decision threshold values for classifying diseased and non-diseased subjects by calculating the sensitivity and specificity[47]. The level of optimal cut-off point (maximum sensitivity while maintaining high specificity) determined by ROC curve analysis is very important because it can provide optimal diagnostic threshold value. Optimal cut-off point in the serum UBAP2 ELISA data in Fig. 7 was 3.033 ng/ml and was lower than that of serum OCN 11.700 ng/ml (Supplementary Fig. 17), indicating that serum UBAP2 has higher sensitivity than serum OCN. In fact, the sensitivity values of UBAP2 and OCN were determined as 86.4% and 50.0%, respectively (Supplementary Fig. 17). Despite the specificity of serum UBAP2 being lower than serum OCN (52.4% vs. 85.7%), higher sensitivity in serum UBAP2 may be an advantage as a biomarker. The AUC value in the ROC curve is used for determining the overall accuracy of target biomarkers for diagnosis. The AUC values of serum UBAP2 and OCN were determined as 0.724 and 0.702, respectively (Supplementary Fig. 17. Because AUC values between 0.7 and 0.8 are considered acceptable discrimination ('good')[48], serum UBAP2 was demonstrated to have good overall accuracy similar to serum OCN. Collectively, although the number of tested subjects was small, the ROC curve analysis results suggest that serum UBAP2 levels may be useful for the diagnosis of osteoporosis as an additional biomarker along with serum OCN and/or other biomarkers. However, to support the potential of UBAP2 as a valuable biomarker, further validation studies using a large number of samples and other cohorts are needed.

This study has some limitations. Firstly, although an intronic rs747091 SNP of *UBAP2* was reported to have a significant association with heel BMD in another ethnic cohort, there is a lack of replication for the analysis of the identified rs2781 SNP in other ethnic populations due to the unavailability of cohorts of women with osteoporosis. Second, despite all efforts, the number of human bone marrow and peripheral blood samples from controls and patients with osteoporosis was relatively small compared with the sample size for the association study. However, we believe that these sample sizes are significant for evaluating the utility of UBAP2. Third, there is no evidence regarding the mechanism by which cytoplasmic UBAP2 could be released in blood plasma, as detected using ELISA tests. Finally, we did not perform a comparative analysis of UBAP2 and bone resorption biomarkers at the protein level in blood plasma samples because the amount of sample obtained from the patients was not sufficient for numerous ELISA experiments.

In conclusion, statistical association studies in two Korean women cohorts and experimental studies in cells, zebrafish, and humans suggest that UBAP2 plays a major role in bone homeostasis through the regulation of bone remodeling.

## Methods

### Ethics statement

This research complies with relevant ethical regulations. The exome-wide association study was approved by the Institutional Review Board (IRB) of the Korea National Institute of Health (approval number: 1041231-170822-BR-062-01). Primary-cultured mouse cell experiments were approved by the Institutional Animal Care and Use Committee (IACUC) of the Ajou University School of Medicine (approval number: IACUC No. 2014-0066) and conducted in accordance with the institutional guidelines established by the Committee. Zebrafish husbandry and animal care were performed in accordance with the guidelines from the Korea Research Institute of Bioscience and Biotechnology (KRIBB) and approved by KRIBB-IACUC (approval number: KRIBB-AEC-21117). Human sample studies were approved by the Ajou University Hospital IRB (approval numbers: AJIRB-GEN-GEN-11-062 and AJIRB-GEN-GEN-11-332), and written informed consent was obtained from all subjects.

### Study design

As the prevalence of osteoporosis is markedly higher in women than in men[6,49], sex was considered in the following study designs. For the exome-wide association study, the sex of the subjects was determined based on self-report and further confirmed by SNP genotypes of sex-chromosomes. For the primary-cultured mouse cell experiment, sex was determined based on phenotypic assessments. For the human sample study, the sex of the participants was determined based on self-report.

Overview of the study design is shown in Supplementary Fig. 1. In the first step of this four-step research, an exome-wide association study was conducted using 6,485 whole-genome exonic SNPs in a total of 1295 subjects from the Korean cohort of women (Ansung). Significance was set below the Bonferroni-corrected level ($p < 7.7 \times 10^{-6}$) in the exome-wide association study. Replication analysis of the discovered significant SNPs was performed in a total of 1371 subjects of another Korean cohort of women (Ansan). In the second step, functional analysis of the target gene (*UBAP2*) corresponding to the identified SNP (rs2781) was performed using in vitro experiments. Viral-vector constructs for knockdown via RNA interference and overexpression of the target gene (*Ubap2*) were transfected into mouse osteoblastic MC3T3-E1 cells and osteoclastic-lineage primary-cultured monocytes, and then gene expression changes of bone remodeling-related proteins involved in osteoblast and osteoclast differentiation were assessed. In the third step, the functional role of the target gene was examined in vivo in an animal model. In zebrafish, expression distribution of the target gene (*ubap2*) during development was observed using mRNA in situ hybridization. Knockdown of the target gene (*ubap2*) in zebrafish

was conducted via injecting MO RNAs into zebrafish embryos and then observing phenotypic alterations in bone and cartilage during development using ARS and Alcian blue staining. In the fourth step, the evaluation of *UBAP2* as a potential osteoporosis biomarker was conducted using human peripheral blood and bone marrow samples from women volunteers with osteoporosis and controls (post-menopausal women). Comparison analysis of mRNA expression of *UBAP2* and bone remodeling-related genes was conducted. Finally, levels of UBAP2 and OCN were compared between controls and women with osteoporosis in blood plasma using ELISA.

## Study cohorts for the association study

The database comprising subject information and SNP genotype information of 8,840 Korean subjects has been generated previously in the Korean Association Resource (KARE) project of the Korean Genome and Epidemiology Study (KoGES)[24]. Briefly, a total of 8840 participants aged 40 to 69 years were recruited from the Ansung (rural area) and Ansan (urban area) cohorts, which represent rural and urban communities in Korea, respectively. The baseline survey of the KoGES involved a total of 10,038 adults from 2001 to 2002, surveyed biannually. The participant survey involved not only their parents' health history and anthropometric data such as height, weight, and waist and hip circumference, but also education and income, physical activity, dietary intake, and so on. This study used the dataset of 2666 women from KARE project. The Ansung cohort, consisting of 312 women with osteoporosis and 983 controls, was used as the discovery data set for exonic SNP association study. The Ansan cohort, consisting of 131 women with osteoporosis and 1240 controls, was used as the validation data set for the replication study. Written informed consent was obtained from all participants, and this research project was approved by the institutional review board of the Korea National Institute of Health. Basic characteristics of the study subjects are described in Supplementary Table 1.

## BMD measurement

BMD was used as a proxy measure for bone strength and resistance to fracture. Although BMD measurement by dual-energy X-ray absorptiometry (DXA) is the standard procedure for evaluating bone quality status, T-score of SOS measured by quantitative ultrasound (QUS) has been used in large population cohort studies because a strong correlation between QUS and DXA and QUS and BMD has been reported[50,51]. The T-score was calculated by dividing the difference between the measured SOS and the mean SOS in a healthy adult population by the standard deviation (SD) of SOS in an adult population. Bone SOS was measured via quantitative ultrasound at the DR and MT, using the Omnisense 7000 P quantitative ultrasound device (Sunlight Medical Ltd, Tel-Aviv, Israel)[52]. Subjects whose T-scores in either DR-SOS or MT-SOS were below −2.6 SD and −3.0 SD, respectively, were considered as patients with osteoporosis, according to the diagnostic categories established for adult women[53]. Subjects whose T-scores in both DR-SOS and MT-SOS were above −1.4 SD and −1.6 SD, respectively, were classified as controls[54].

## SNP genotyping and quality control

Genotyping data were provided by the Center for Genome Science, the Korea National Institute of Health. The detailed genotyping and quality control processes were described previously[24]. Briefly, DNA samples were isolated from the peripheral blood of participants and subjected to SNP genotyping using the Affymetrix Genome-Wide Human SNP array 5.0 (Affymetrix; Santa Clara, CA, USA). The accuracy of the genotyping was calculated through Bayesian Robust Linear Modeling using the Mahalanobis Distance genotyping algorithm[4]. Samples showing lower genotyping accuracies (≤98%), high missing genotype call rates (≥4%), high heterozygosity (>30%), or gender biases were excluded from this study.

## Selection of whole-genome exonic SNPs

We selected 6485 whole-genome exonic SNPs from the QC-filtered 352,228 SNPs, based on the information in dbSNP v.129. A total of 6983 SNPs linked to 4469 genes were extracted as 'an exonic category', including nonsynonymous (1262), synonymous (1493), frameshift (3), stop gained (8), stop lost (3), splicing site (290), 5′-UTR (549), 3′-UTR (3,374), and mature miRNA (1) SNPs, and 498 SNPs were overlapping using BioMart version 5.0[25].

## SNP selection and imputation the *UBAP2* region

We selected 17 genotyped SNPs in *UBAP2* based on their location within the gene boundary (5 kb upstream and downstream of the first and last exons) according to NCBI human genome build 36 (Supplementary Table 3). We also selected 79 imputed SNPs in *UBAP2* via genotype imputation analysis using MACH 1.0.16. The Han Chinese from Beijing (CHB) and Japanese in Tokyo (JPT), obtained from the Phase II HapMap database (release 24), were used as references (Supplementary Table 3). The imputed SNPs with a minor allele frequency <0.01 or coefficient of determination values < 0.5 were excluded.

## Cell culture and induction of osteoblastogenesis and osteoclastogenesis

HEK293TN cell line (System Biosciences, Palo Alto, CA, USA), the production of lentiviral particles, and HeLa cell line (Korean Cell Line Bank, Seoul, Korea) were grown in Dulbecco's Modified Eagle Medium (Thermo Fisher Scientific Inc., Waltham, MA, USA) supplemented with fetal bovine serum (10%; Sigma-Aldrich, St. Louis, MO, USA), penicillin (100 U/ml; Duchefa, RV Haarlem, Netherlands), and streptomycin (100 μg/ml; Duchefa), and incubated in a humidified atmosphere at 37 °C under 5% $CO_2$.

Mouse pre-osteoblast MC3T3-E1 cells (RIKEN Cell Bank, Tsukuba, Japan) and primary monocytes isolated from mouse bone marrow were grown in α-MEM medium (Life Technology; Carlsbad, CA, USA) supplemented with fetal bovine serum (10%; Sigma-Aldrich), penicillin (100 U/ml; Duchefa), and streptomycin (100 μg/ml; Duchefa), and incubated in a humidified atmosphere at 37 °C and $CO_2$ (5%). Two types of cells, before induction of differentiation, were infected with sh*Ubap2* and Ubap2 viral particles for 24 h to regulate gene expression and selected using puromycin (4 μg/ml). MC3T3-E1 cells at passages 5–10 (after purchase) were used for all experiments. For inducing osteoblastogenesis, MC3T3-E1 cells were cultured in the same medium, supplemented with 50 μg/ml ascorbic acid and 10 mM β-glycerophosphate for 5–14 days, and the medium was changed every 3 days. To prepare primary-cultured monocytes, bone marrows from the femoral bones of 6-week-old female *BALB/c* mice were flushed into α-MEM medium in the presence of 1 mM ascorbate-2-phosphate (Sigma-Aldrich). The primary-cultured monocytes were validated via immunophenotypic analysis with a CD11b antibody (BioLegend; San Diego, CA, USA) using the FACS Aria III cell sorter (BD Biosciences; San Jose, CA, USA) and FACS Diva software (BD Biosciences) (Supplementary Fig. 6). For the induction of osteoclastogenesis, primary monocytes were cultured in the presence of 30 ng/ml macrophage colony-stimulating factor (M-CSF) (PeproTech; Rocky Hill, NJ, USA) and 50 ng/ml RANKL (PeproTech) for 4 days[55]. All efforts were made to minimize animal suffering and reduce the number of mice used.

## ALP assay and staining and mineralized nodule formation

For the measurement of ALP activity, the cultured cells were lysed using an extraction solution of the TRACP & ALP Assay Kit (Takara Bio Inc.; Shiga, Japan) and incubated overnight at 4 °C after washing twice with saline. The lysates were reacted with p-nitrophenylphosphate (5 mM) as a substrate for 30 min at 37 °C, and the reaction was stopped by adding NaOH (0.5 N). The absorbance was measured at 405 nm

(Bio-Rad Laboratories; Hercules, CA, USA). For ALP staining, the cultured cells were fixed by drying for 10 min, after washing twice with saline, and incubated with BCIP/NBT (Sigma-Aldrich) for 30 min in the dark. Mineralization of MC3T3-E1 cells was assessed via ARS staining (40 mM; Sigma-Aldrich) after fixation in ethanol (70%). ARS was extracted using cetylpyridinium chloride (10%) to quantify the mineralized nodule, and the absorbance of the supernatants was measured at 595 nm.

### TRAP assay and staining
Differentiated monocyte cells were subjected to TRAP activity assay and staining using TRACP & ALP Assay Kit (Takara Bio Inc.) and Acid-Phosphatase Kit (Sigma-Aldrich) as per the manufacturer's instructions. TRAP activity was measured at 405 nm (Bio-Rad Laboratories).

### Preparation of lentiviral and retroviral constructs
Short hairpin RNAs (shRNAs) against mouse *Ubap2* (sh*Ubap2*_#1: TRCN0000241149 and sh*Ubap2*_#2: TRCN0000241151), mouse *Cdh1* (sh*Cdh1*_#1: TRCN0000042578, sh*Cdh1*_#2: TRCN0000042579, sh*Cdh1*_#3: TRCN0000042580, sh*Cdh1*_#4: TRCN0000042581, and sh*Cdh1*_#5: TRCN0000042582), and mouse *Fosl1* (sh*Fosl1*_#1: TRCN0000042684, sh*Fosl1*_#2: TRCN0000042685, sh*Fosl1*_#3: TRCN0000042687, sh*Fosl1*_#4: TRCN0000310960, and sh*Fosl1*_#5: TRCN0000316021) cloned into the pLKO.1-puro vector were purchased from Sigma-Aldrich (Supplementary Table 7). For *Ubap2* overexpression, mouse *Ubap2* cDNA was amplified from mouse testis cDNA library using PCR and cloned into the pDON-5 Neo vector (Takara Bio Inc.). shRNAs for *Ubap2*, *Cdh1*, and *Fosl1* and *Ubap2* overexpression constructs were transfected using pPACKH1™ Lentivector Packaging kit (System Biosciences) into 293TN cells for 72 h using Lipofectamine 3000 (Thermo Fisher Scientific Inc.) and then the collected supernatants were used to treat MC3T3-E1 cells or primary monocytes. pLKO.1-puro empty vector and pDON-5 Neo empty vector were used as negative controls.

### Quantitative reverse-transcription PCR (qRT-PCR)
Total RNAs from mouse pre-osteoblast MC3T3-E1 cells, mouse monocytes, and human samples were isolated using TRIzol reagent (Invitrogen; Carlsbad, CA, USA) and treated with RNase-free DNase I (Invitrogen). One microgram of total RNA was subsequently reverse-transcribed using the RevertAid™ Minus First Strand cDNA Synthesis Kit (Fermentas Inc.; Hanover, MD, USA) with both the oligo (dT) 15-18 primer and the random hexamer primer. The qRT-PCR was performed using the Qiagen Rotor-Q (Qiagen; Hilden, Germany) or CFX Connect™ Real-Time PCR Detection System (Bio-Rad Laboratories). PCR amplification (40 cycles) was performed with reaction mixtures of a total volume of 10 μl containing cDNA (100 ng) using the SYBR Green I qPCR kit (Takara Bio Inc.) according to the manufacturer's instructions. Gene-specific primers (COSMOGENETECH, Seoul, Korea) used for qRT-PCR are shown in Supplementary Table 8.

Relative quantification of gene expression was performed using the cycle threshold (Ct) method as described by the manufacturer's manual (Qiagen or Bio-Rad) and normalized to mouse *Gusb* or human *GUSB* expression. Relative gene expression of target genes was calculated as $2^{-delta (\Delta) Ct}$ ($\Delta Ct = Ct_{target gene} - Ct_{control gene}$) and fold change was calculated as $2^{-\Delta\Delta Ct}$ ($\Delta\Delta Ct = \Delta Ct_{target sample} - \Delta Ct_{reference sample}$). The quantification value was expressed as the fold change relative to the control.

### Western blotting
MC3T3-E1 cells and primary monocytes were lysed in RIPA buffer (150 mM NaCl, 1% Nonidet P-40, 0.5% sodium deoxycholate, 0.1% SDS, and 50 mM of Tris buffer, pH 8.0). Protein concentration was determined using the Dc Protein assay kit (Bio-Rad Laboratories). Protein samples were analyzed via SDS-PAGE on polyacrylamide gels (12%).

Proteins were electroblotted onto PVDF membrane (Millipore; Concord Road Billerica, MA, USA). The membrane blots were blocked with BSA (5%, w/v), incubated with primary and secondary antibodies, and then visualized using the WEST-ZOL plus ECL Western blotting detection system (iNtRON Biotechnology; Daejeon, Korea). Anti-Ubap2 antibodies were purchased from Abcepta (1:500; AP12773a; San Diego, CA, USA), Abcam (1:1000; ab197083; Cambridge, UK), and Bethyl Laboratories, Inc. (1:1000; A304-626A; Montgomery, TX, USA). Anti-E-cadherin antibodies (1:1000; 20874-1-AP) were from Proteintech (Rosemnot, IL, USA), and anti-Fra1 (1:1000; sc-28310) and anti-β-actin antibodies (1:2500; sc-47778) were from Santa Cruz Biotechnology (Santa Cruz, CA, USA). HRP-conjugated goat anti-rabbit IgG and HRP-conjugated goat anti-mouse IgG antibodies were purchased from Bethyl Laboratories, Inc.

### Whole-mount in situ RNA hybridization in zebrafish
Zebrafish were raised at 28.5 °C with a life cycle of 14 h light/10 h dark. Fertilized eggs were collected from group-mated wild type AB, and embryos were staged as previously described[56]. Full-length cDNA clones for zebrafish *ubap2a* and *ubap2b* were purchased from Thermo Fisher Scientific Inc. The PCR fragment spanning the middle 1-Kb of the clone to synthesize an RNA probe for the whole-mount in situ RNA hybridization was amplified using the primer pairs for *ubap2a* (5′-AGG GGC CAA TGA CAC TAC AG-3′ and 5′-AGG GCA GAT TCT GAA CCA AA-3′) and *ubap2b* (5′-CGG AGA GGA CTT TTC TAT TTT G-3′ and 5′-CGT GTT AGA GGC AGG ACA GAG CG-3′) designing the appropriate PCR cycles, followed by subcloning into the pCRII TOPO vector (Invitrogen). Digoxigenin (DIG)-labeled anti-sense and sense RNAs were prepared using DIG RNA labeling kit (Roche; Basel, Switzerland) after linearization of the DNA template. After embryos were fixed with paraformaldehyde (4%), whole-mount in situ RNA hybridization was performed as previously reported[57]. After an extensive wash with saline-sodium citrate buffer, the hybridized RNA probes were detected via incubation with ALP-conjugated anti-DIG Fab antibody (Roche), followed by visualization with NBT/BCIP as an ALP substrate.

### Whole-mount immunostaining in Zebrafish
Whole-mount immunostaining of *Ubap2* in the zebrafish larvae was performed as previously described[58]. Briefly, 3 dpf larvae were fixed in 4% paraformaldehyde diluted in 1 × PBS with 0.25% Triton X (PBST) at 4 °C overnight. After PBST washing, antigens retrieval was performed using TrisHCl (pH 9.0, 150 mM) at 70 °C for 20 min. The larvae were washed with PBST and then treated with 0.05% trypsin-EDTA for 45 min on ice, and incubated with a blocking buffer (2% normal goat serum, 1% bovine serum albumin, and 1% dimethyl sulfoxide) for 1 h. The larvae were incubated overnight at 4 °C with the primary antibody (Ubap2, 1:10, Abcepta, AP12773a) or without the primary antibody as a negative control and then incubated with a secondary antibody conjugated with Alexa fluorophore 488 (1:500, Invitrogen). Larvae were mounted in 1% low melting agarose and imaged with a confocal microscope (FV1000 confocal microscope, Olympus)[58].

### MO RNA injection and efficacy validation in zebrafish
*ubap2* knockdown in zebrafish was performed by injecting the synthesized MO RNAs against *ubap2a* or *Ubap2b* into the embryo. The genomic structures of zebrafish *ubap2* were predicted in the annotated zebrafish genome (assembly version Zv9 and GRCz11) in the Ensemble database (www.ensembl.org). MO oligonucleotides (*ubap2a* splicing blocking e3i3: 5′-CCG AAG AGA GCT TCT TGT ACC TGT T-3′; *ubap2a* AUG translation blocking: 5′- AAA AAT CTG CTA ACA CGC TTC ACT C; *ubap2b* splicing blocking: e4i4 5′- TGT ATA AAG AAC ACA CCT TGA CAG T-3′; *ubap2b* splicing blocking: i4e5 5′- GAG TCC TGT GCA ATC AAA GCA GCA C-3′; and standard control MO: 5′- CCT CTT ACC TCA GTT ACA ATT TAT A-3′) were purchased from Gene Tools [http://www.genetools.com]. Fertilized wild-type eggs at one- or two-cell

stages were microinjected with *ubap2*-MOs using a microinjector (PV830 Pneumatic picopump) (World Precision Instruments Ltd.; Hitchin, Hertfordshire, UK) and fixed at desired stages for further experiments. For testing the efficacy of the *ubap2a and ubap2b* knockdown, total RNA was extracted from 30 h postfertilization (hpf) embryos using TRIzol reagent (Invitrogen). cDNA synthesis and qRT-PCR were performed as described above. Primer sequences for *ubap2a* were as follows: 5′-GCT GCC CAC TAC CAC ACA G-3′ and 5′-TGC TGC TTT CCC TCT CTT TC-3′ for exons 2 and 5 of *ubap2a*, respectively. For *ubap2b*, primers 5′-GCT CAT GGA GGT TAC AGG CA-3′ and 5′-CAG TCG CAA CCA GGA AGG TGT-3′ were used. PCR cycle conditions were 94 °C for 2 min, 35 cycles (94 °C for 30 s; 55 °C for 30 s, 72 °C for 45 s), and 72 °C for 5 min. The results were visualized by running the samples on agarose gel for detecting aberrant splicing bands. These semi-qRT-PCR primers used in Supplementary Fig. 11g–i covered two exons for checking the correct PCR fragment sizes; they were not tested for their efficiency. A full-length cDNA clone for zebrafish *ubap2a* was purchased from Open Biosystems (Clone ID: 6790255; Huntsville, AL, USA). *ubap2b* cDNA was amplified using a primer pair (forward primer 5′-CGG AGA GGA CTT TTC TAT TTT G-3′; reverse primer 5′-CGT GTT AGA GGC AGG ACA GAG CG-3′) with a PCR cycle [94 °C, 5 min, 30 cycles of (94 °C, 30 s; 55 °C, 30 s; 72 °C, 3 min), 72 °C, 7 min], followed by subcloning into the pCRII-TOPO vector (Thermo Fisher Scientific Inc.). Full-length cDNAs of *ubap2a* and *ubap2b* were subcloned into a pCS2+ vector, and mRNAs were synthesized using the mMESSAGE mMACHINE™ SP6 Transcription Kit (Invitrogen).

### ARS and Alcian blue staining in zebrafish

ARS staining of zebrafish was performed as previously published[59]. Briefly, fixed embryos were preincubated in KOH (1%) for 1 h and then incubated in ARS (0.003%) in KOH (1%) overnight. The next day, stained embryos were cleared with KOH (1%), stored in glycerol (100%), and imaged. Alcian blue staining was performed as previously published[60]. Briefly, fixed embryos were stained with Alcian blue solution (1% HCl, 70% ethanol, 1% (W/V) Alcian blue) overnight, followed by incubation with Alcian blue clearing solution (5% HCl, 70% ethanol) and dehydration with ethanol series (25, 50, 75, and 100% ethanol). Processed embryos were stored in glycerol (100%) and imaged using Olympus SZX16 stereoscope attached with Olympus XC10 CCD color camera.

### Recruitment of human subjects for bone marrow and peripheral blood collection

To investigate *UBAP2* expression in humans, we recruited 45 women to collect bone marrow samples (15 controls and 30 patients with osteoporosis) (Supplementary Table 4) and 63 subjects to collect peripheral blood (32 controls and 31 patients with osteoporosis) (Supplementary Table 5) at the departments of Endocrinology and Metabolism, Family Practice and Community Health, and Orthopedic Surgery in Ajou University Hospital. All subjects were postmenopausal women aged above 50 years. BMDs (g cm⁻²) at the lumbar spine (L1-L4), femur, and hip were measured using X-ray absorptiometry (GE Lunar; Madison, WI, USA). Osteoporosis was diagnosed according to World Health Organization parameters (a lumbar spine or femoral T-score lower than −2.5 SD). All bone marrow samples were obtained from surgical bones after vertebral fracture or hip fracture surgery. Peripheral whole blood and bone marrows were centrifuged, and buffy coat and plasma layers were separated. Total RNA was isolated using IQeasy™ Plus Blood RNA Extraction Mini Kit (iNtRON Biotechnology) from buffy coat of peripheral blood and bone marrow samples.

### ELISA

For the assessment of UBAP2 and OCN levels in the blood plasma, two ELISA kits were used for UBAP2 (USCN Life Science Technology; Wuhan, China, and antibodies-online GmbH; Aachen, Germany) and

OCN (Merck; Kenilworth, NJ, USA). Briefly, each 100 μl of standard, blank, and sample was added into the appropriate wells and incubated for 2 h at 37 °C. After removing the liquid, detection reagents A and B were added to each well and incubated for 1 h at 37 °C. Stop solution was added and incubated for 20 min and then, the OD of the samples was measured at 450 nm. The protein concentration of samples was calculated by analyzing the ELISA data with respect to the standard curve.

### Statistical analysis

Significant differences in characteristics of the controls and patients with osteoporosis presented in Supplementary Table 1 were determined using two-tailed Student's *t*-test. Statistical analysis of the association study was performed using a whole-genome association analysis toolset PLINK version 1.07 [https://zzz.bwh.harvard.edu/plink/] and PASW Statistics software, version 17.0 (SPSS Inc.; Chicago, IL, USA). To minimize the effects of age differences between osteoporosis and control groups, all logistic regression analyzes were adjusted for age as covariates. The exome-wide association study was performed under the additive genetic model, and *p*-values were adjusted for multiple tests using the Bonferroni-corrected significance level ($p < 7.7 \times 10^{-6}$). Replication association analysis was performed under the additive genetic model and a *p*-value less than 0.05 was considered statistically significant. The LocusZoom internet tools [http://locuszoom.sph.umich.edu/] and the SNP annotation and proxy search database [http://www.broadinstitute.org/mpg/snap/] were used for regional association plot drawing.

In the experimental studies, all the experiments were repeated independently at least three times unless stated otherwise, and the results were presented as the mean ± SD as indicated. Statistical analyses were performed using Prism version 9 (GraphPad; Boston, MA, USA) and statistical significance between the groups was calculated using Student's *t*-test. Results with a *p*-value of less than 0.05 were considered statistically significant. Comparison of multiple groups was performed using one-way or two-way ANOVA, followed by Tukey's honestly significant difference post hoc test for correction of multiple comparisons.

### Reporting summary

Further information on research design is available in the Nature Portfolio Reporting Summary linked to this article.

## Data availability

The exome-wide association study was performed using the dataset (subject information and SNP genotype data) of 2,666 women, which was originally generated in the KARE project (total 8,840 subjects) of KoGES [https://www.nih.go.kr/ko/main/contents.do?menuNo=300563] supported by the Korean National Institute of Health (KNIH), which is an affiliated organization of the Korea Disease Control and Prevention Agency (KDCA). The subject information and SNP genotype data used in this study are owned entirely by the KNIH, and disclosure of the raw data to the public without permission is strictly prohibited. Although the personal raw data of SNP genotype and epidemiology and subject information cannot be publicly shared because of legal restriction imposed by the Korean Bioethics and Safety Act, these data (with anonymization) can be used for research purpose. In principle, the raw data of subject information and SNP genotype used in this study are available with permission from the Institutional Review Board of KNIH [https://www.nih.go.kr/eng/main/main.do] for researchers in Korea who meet confidential data access criteria. These data can also be available for researchers overseas when undertaking an international cooperative research project and when the KDCA approves it. To access more information for data distribution, please refer to the Korea Biobank Project [https://www.kdca.go.kr/contents.es?mid=a30326000000]. All data supporting the findings described in this manuscript are available in

the article and in the Supplementary Information and from the corresponding author upon request. Source data are provided with this paper.

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

## Acknowledgements

This study was supported by a grant from the Korea Health Technology R&D Project through the Korea Health Industry Development Institution (KHIDI), funded by the Ministry of Health & Welfare, Republic of Korea (grant number: HR22C1734) (S-Y.J.), the Basic Science Research Program through the National Research Foundation of Korea (NRF) funded by the Ministry of Education (NRF-2018R1D1A1B07048040) (J.K.), and KRIBB Research Initiative Program (KGM9992211) (J-S.L.). Jeonghyun Kim, Bo-Young Kim, Jeong-Soo Lee, and Yun-Mi Jeong contributed equally to this work.

## Author contributions

S-Y.J., Y-S.C., and H-S.J. are the project leaders and participated in the study design, data analysis, and manuscript writing and editing. H-S.J. and S-S.K. designed the genetic association study and analyzed the data. J.K. and B-Y.K. performed in vitro cell experiments and mRNA and protein level comparison experiments in human samples. J-S.L., Y-M.J., and H-J.C. performed zebrafish experiments, and analyzed and interpreted the results. D.K. performed Ingenuity Pathway Analysis. E.P. prepared and cultured monocytes from bone marrows. B-T.K., Y.J.C., Y-Y.W., and Y-S.C. recruited human subjects, prepared peripheral blood and bone marrow samples, and obtained clinical data. J.K., B-Y.K., J-S.L., H-S.J., Y-S.C., and S-Y.J. wrote the manuscript with input from all authors. All of the authors reviewed the manuscript.

## Competing interests

The authors declare no competing interests.
