## [Peer Review File · Nature Communications]

UBAP2 plays a role in bone homeostasis through the regulation of osteoblastogenesis and osteoclastogenesisREVIEWER COMMENTS

Reviewer #1 (Remarks to the Author):

The paper entitled “UBAP2 as a Novel Biomarker for the Clinical Assessment of Osteoporosis in Women” is potentially interesting. The authors found that rs2781 single nucleotide polymorphism in UBAP2 was significantly associated with osteoporosis and bone density with genome-wide significance level. Knockdown of ubap2 in zebrafish caused abnormal osteoblastogenesis and osteoclastogenesis and abnormal bone formation, respectively. UBAP2 mRNA levels were significantly reduced in bone marrows but upregulated in peripheral bloods from women with osteoporosis.

There are some issues

UBAP2 expression level was found closely correlated with the blood plasma level of the representative osteoporosis biomarker osteocalcin. Ubiquitin-associated protein 2 is usually localised in nuclear or cytoplasmic, and is not sure how it was released into circulation? UBAP2 ELISA specificity might need to be verified ?

Where is Ubiquitin-associated protein 2 localized in osteoclasts and osteoblasts?

Figure 1, 2 and Figure S7, it is important to show the effect of shRNA on the percentage level of UBAP2 protein knockdown by western blot, and if UBAP2 protein expression is secreted in supernatants, hence in blood plasma?

Figure 3, it was not clear if osteoclasts were affected in ubap2 KD zebrafish? TRACP staining might be performed. In addition, Alizarin red staining results are best quantified ? KO of ubap2 is to be confirmed by western blot?

In the abstract, there was statement referring to knockdown of Ubap2 in mouse? Which was not shown?

Reviewer #2 (Remarks to the Author):

The authors first describe a 2 stage case-control GWAS of osteoporosis in Korean women using very small discovery (312 cases, and 983 controls) and replication (131 cases, and 1,240 control) samples. DNA samples were genotyped on Affymetrix 500K array, however, only 6,485 exonic SNPs were assessed, which seems unusually given that GWAS associations are frequently detected in non-coding regions. The authors define the significance threshold at $p < 7.7E-6$ (Bonferroni), however, none of the SNPs in the discovery study achieved significance or were suggestive (e.g. best SNP rs2781 $p = 2.3E-4$), therefore strictly speaking, there was no significant association to replicate. Subsequently, the authors combined the two cohorts and repeated the GWAS. There are several concerns about the statistical approach; a 2-stage GWAS with no significant findings followed by a combined analysis (n.b. this was not described as a meta-analysis and no covariate appears to have been included to represent the different populations combined in the analysis), therefore, the report effectively presents a very small exonic GWAS of 443 cases 2,223 controls, with one significant finding, rs2781 ($p = 6.1E-7$), without replication. Age was used as a covariate, however, some measure of body size should also have been included such as weight, height or body mass index; indeed both adult body size and BMI have both been shown through GWAS to be associated with SNPs in UBAP2 (rs544957562 and rs7019441 respectively). Principal component analysis (PCA) is a routinely used to correct for population stratification in GWAS and to cluster individuals by ancestry, however, this does not appear to have been done in this study. It is also not stated if any of the participants were related, how this was assessed and if necessary, how this was accounted for in the analyses. It is not clear what modality (or manufacturer) was used to classify the cases as osteoporotic – DXA, QUS, QCT? The authors describe an association with UBAP2 as novel, but an association between quantitative ultrasound measures of bone density has previously been reported with rs747091 in UBAP2 (Morris et al. Nat Genet. 2019 Feb;51(2):258-266.); rs2781 is in linkage disequilibrium $D' = -0.85$, $r^2 = 0.61$, suggesting that the finding may represent the same association signal as previously reported. The current report would be of particular value if the functional SNP had been confidently identified, however, that cannot be determined from the information available (but seems unlikely given the statistical power of the Morris et al 2019 study).

The authors next performed a GWAS of the quantitative ultrasound (QUS) phenotype SOS. They refer to bone density; bone density measurements are based on X-ray attenuation (either QCT or DXA), whereas ultrasound parameters reflect the structural anisotropy of bone; bone properties other than density are detectable by QUS and not by DXA (e.g. elasticity and microstructure). QUS measures can be manipulated to estimate BMD (eBMD), but it is not clear that was done in this study. If the osteoporosis cases were originally defined using SOS (and not DXA), then the quantitative trait GWAS seems like duplication and it is unsurprising that it also highlighted the same genetic variant. GWAS in the literature do tend to be Caucasian-centric and a GWAS of osteoporosis in women of non-European ancestry is potentially of substantial merit and interest.

The reference that is provided to the clinical details of the GWAS cohorts (39) is not adequate and more detail is needed in the text and supplement of this paper. Also, Line 132 and table 1 state - cases, 443; controls, 2,223 were used in the GWAS, yet the abstract states the GWAS was in 3,569 women, so this cannot be reconciled.

A series of functional analyses of the target gene (UBAP2) were performed using in vitro experiments. These appear well done and provide interesting data on the potential role of UBAP2 in bone cells.

The statement is made that "UBAP2 has great potential as a novel bone biomarker for the clinical assessment of osteoporosis in women, including diagnosis and evaluation of the effects of drug treatments" – based on the evidence provided this is a substantial overstatement and needs to be revised e.g. "has great" versus "may have". Furthermore, ROC curves are frequently used to show sensitivity and specificity for a clinical test. The area under the ROC curve gives an idea about the benefit of using the test; although the number of available subjects is small, this could be examined in regard to the utility of UBAP2 as a clinical biomarker for bone and in comparison to osteocalcin.

Some of the text in the discussion is repetition of the results and more discussion is needed on the relative effect size and minor allele frequency of this variant compared to other variants for genetics of osteoporosis, pleiotropic associations seen with UBAP2 and other traits such as body size and BMI, and the biology of UBAP2.

An important element that is missing from the report is the investigation of the different effects of the C-allele at rs2871 versus the G-allele in the functional studies.

Reviewer #3 (Remarks to the Author):

The manuscript "UBAP2 as a novel biomarker for the clinical assessment of osteoporosis in woman" presents convincing data that UBAP2, a protein likely involved in the ubiquitination pathway, has bone anabolic, but also pro-osteoclastogenic functions, and because of its increased concentration in peripheral blood samples might be a useful biomarker to detect osteoporosis. The authors show that knockdown of UBAP2 results in bone mineralization defects in MC3T3-E1 osteoblast cell culture, and that osteoclastogenesis is enhanced after knockdown in primary-cultured monocytes. They investigate the function of ubap2 in zebrafish by using a morpholino knockdown approach and combine their in vitro and in vivo results with the investigation of UBAP2 levels in bone marrow and peripheral blood samples of patients. To my knowledge, a role in bone homeostasis and bone pathology has not been demonstrated for UBAP2 before, thus the results presented are novel and definitely interesting.

However, some questions remain:

1) The function of upab2 in formation and calcification of bone and cartilage in zebrafish remains somewhat enigmatic. Alcian Blue staining and Alizarin Red staining suggest reduced formation. However, certain improvements should be made to confirm this finding. Because of a genome duplication event in teleost fish species, many genes have two paralogues and according to ZFIN (zebrafish information network), this also seems to be true for upab2. Which gene (ubap2a or ubap2b) are we dealing with here? Testing of both paralogs may be necessary or an explanation should be provided. Also, morpholino knockdown approaches have their caveats, and have to make use of stringent controls, to exclude developmental delay phenotypes. Importantly, rather than uninjected controls, control-morpholino-injected embryos should be used (e.g. mismatch morpholino, see Stainier et al. 2017 PlosGenetics). In addition, a second morpholino (e.g. translation blocking morpholino) or alternatively a mutant is required to validate the data, as morpholino usage often produces off-target effects. While I do not doubt the results presented with the splice blocking morpholino, I suggest to add a second morpholino and to control for unspecific effects by using a mismatch morpholino as a control. If the same effects are seen, the phenotype should be rescued by injection of ubap2a/b mRNA. Alternatively, mutant zebrafish, created by the Sanger institute, or by injection of ubap2a/b targeting sgRNAs and Cas9 should be analyzed.

2) The authors discuss a potential function of Anxa2 downstream of UBAP2. In order to provide mechanistic insight into UBAP2's action I suggest to investigate the hypothesis of Anxa2 in cell culture assays. For example, Anxa2 protein and/or transcript levels could be evaluated after UBAP2 knockdown, and reduced matrix formation as well as increased osteoclast formation could be rescued with overexpression of Anxa2.

Furthermore I have some minor comments:

3)line 140 onwards. The sentence is a little hard to understand, maybe the content can be split to two sentences?

4) line 214 onwards. I am afraid I do not fully understand the meaning of this sentence (calcific deposition per se). Can the authors please rephrase the sentence?

5) please use bglap or ocn throughout the text (mentioning the alternative name once when the gene is mentioned first).

6) Please clarify for the readers whether OCN levels are up or down (in previous studies of osteoporotic women).

7) line 265. Numerous genetic risk factors are responsible... (add the word 'are')

8) line 282 and 283. "We also..." used twice, maybe replace?

9) line 286. hematopoietic, instead of hemopoietic

10) please use the term bone mineral density instead of bone density

11) line 497. Were the primers used for qRT-PCR exon spanning? Was efficiency testing performed?

12) line 606. The experiments were repeated independently at least three times... Are the results shown representative results of one experiment or the average of three experiments?

13) The figure legends contain a lot of material and methods information (e.g. the cells were incubated with 10% cetylpyridium chloride at room temperature). If this is not a journal requirement, please move this sort of information to the material & methods section.

14) line 799. All experiments were repeated three times with duplicated analyses. - What does duplicated analyses exactly mean?

15) Fig. 5 and Fig. 6 - aren't controls also patients? maybe the legend could compare healthy versus osteoporotic or alike?

16) Reporting summary: isn't the human data used clinical data?

17) It seems there are no FACS gating plots - does this section need to be answered (e.g. on contour lines)? If so, the plots would need to be shown.

Response to Reviewers' comments

We thank the reviewers for their valuable comments on our manuscript. We have revised the manuscript according to their comments and supplied detailed, point-by-point responses below. All changes in the text appear in red. The English language of the manuscript was proofread and edited by native-English-speaking, professional scientists at EDITAGE, a scientific research paper editing company.

Reviewer #1 (Remarks to the Author):

The paper entitled “UBAP2 as a Novel Biomarker for the Clinical Assessment of Osteoporosis in Women” is potentially interesting. The authors found that rs2781 single nucleotide polymorphism in UBAP2 was significantly associated with osteoporosis and bone density with genome-wide significance level. Knockdown of ubap2 in zebrafish caused abnormal osteoblastogenesis and osteoclastogenesis and abnormal bone formation, respectively. UBAP2 mRNA levels were significantly reduced in bone marrows but upregulated in peripheral bloods from women with osteoporosis.

There are some issues

· UBAP2 expression level was found closely correlated with the blood plasma level of the representative osteoporosis biomarker osteocalcin. Ubiquitin-associated protein 2 is usually localized in nuclear or cytoplasmic, and is not sure how it was released into circulation? UBAP2 ELISA specificity might need to be verified? and if UBAP2 protein expression is secreted in supernatants, hence in blood plasma?

Where is Ubiquitin-associated protein 2 localized in osteoclasts and osteoblasts?

We thank the reviewer for pointing this out. We addressed this in the following manner.

→ The Human Protein Atlas data (www.proteinatlas.org) indicate that UBAP2 is a cytosolic protein.

HUMAN PROTEIN ATLAS SUMMARY ¹	
Protein ¹	Ubiquitin associated protein 2
Gene name ¹	UBAP2 (bA176F3.5, FLJ22435, KIAA1491)
Tissue specificity ¹	Low tissue specificity
Tissue expression cluster ¹	Testis - Cell cycle regulation (mainly)
Single cell type specificity ¹	Low cell type specificity
Single cell type expression cluster ¹	Non-specific - Transcription (mainly)
Immune cell specificity ¹	Low immune cell specificity
Brain specificity ¹	Low human brain regional specificity
Cancer prognostic summary	Prognostic marker in liver cancer (unfavorable), renal cancer (favorable) and ovarian cancer (favorable)
Predicted location ¹	Intracellular
Subcellular summary ¹	Located in Cytosol
Gene summary (Entrez) ²	The protein encoded by this gene contains a UBA (ubiquitin associated) domain, which is characteristic of proteins that function in the ubiquitination pathway. This gene may show increased expression in the adrenal gland and lymphatic tissues. Alternative splicing results in multiple transcript variants. [provided by RefSeq, Sep 2013] show less

→ To determine the localization of Uba2, we performed immunocytochemistry of endogenous Uba2 in osteoblasts and osteoclasts. The data revealed that Uba2 was predominantly expressed in cytosol and also found in the culture media of pre-osteoblast MC3T3-E1 cells. Moreover, its level was increased by induction of osteoblastic differentiation (Supplementary Fig. S7). These results suggest that the Uba2 protein may be present both intracellularly and extracellularly, and the protein may circulate in blood through extracellular secretion, similar to osteocalcin. (Please refer to Fig. S7)

→ ROC curve regression analysis was performed using R program of the pROC package (version 1.18) (Reference 27. Robin et al. BMC Bioinformatics 2011; 12, 77. DOI:10.1186/1471-2105-12-77) with the ELISA value data of UBAP2 and OCN in Fig. 7. The optimal cut-off point, specificity, sensitivity, and the area under the curve (AUC) were determined. The results are shown in Supplementary Fig. S15 and the detailed explanation and interpretation of the results are added in the Results (lines 321-332) and Discussion sections (lines 458- 477).

· Figure 1, 2 and Figure S7, it is important to show the effect of shRNA on the percentage level of UBAP2 protein knockdown by western blot.

→ According to the reviewer's recommendation, the knocked-down level of Ubp2 protein by *Ubp2* shRNA was quantitatively analyzed using Image Processing and Analysis in Java (Image J) software (<http://imagej.nih.gov/ij/>), and the intensity was normalized to β -actin. The sh*Ubp2*_#1 clone showed more than 90% knockdown-efficiency in MC3T3-E1 cells (Please refer to Fig. S8b).

· Figure 3, it was not clear if osteoclasts were affected in *ubap2* KD zebrafish? TRACP staining might be performed.

→ We thank the reviewer for pointing out this concern. It is known that osteoclastogenesis in zebrafish occurs after 20 dpf (days postfertilization) (Witten et al., J. Morphol. 2001; 250(3):197-207. DOI: 10.1002/jmor.1065). It must be noted that all morpholino RNA experiments were performed within 6 dpf because of the limited time of the morpholino RNA effects in zebrafish. Because it is quite early in the developmental stages before osteoclastogenesis begins, it was not possible to test the TRAP staining for osteoblast formation.

· In the abstract, there was statement referring to knockdown of *Ubp2* in mouse? Which was not shown.

→ We thank the reviewer for pointing it out. We have changed it to “knockdown of *Ubp2* in cultured cells” in the abstract, line 65.

· In addition, Alizarin red staining results are best quantified?

→ According to the reviewer's recommendation, we conducted quantification of the Alizarin-red staining in zebrafish. In the revised manuscript, two zebrafish *ubap2* paralogs, namely *ubap2a* (*ubap2* in the previous manuscript) and *ubap2b* were tested. We performed zebrafish knockdown experiments using two distinct morpholino RNAs against *ubap2a* and *ubap2b* (*ubap2a*-AUG MO and *ubap2a*-e3i3 MO for *ubap2a* and *ubap2b*-e4i4 MO and *ubap2a*-i4e5 MO for *ubap2b*). The lengths of the notochords in the larvae at 6 dpf were measured and quantified with statistical analysis. The results are shown in Fig. 3 and Supplementary Fig. S11.

· KO of *ubap2* is to be confirmed by western blot?

→ We thank the reviewer for the kind suggestion. Because *ubap2a*-e3i3 MO, *ubap2b*-e4i4 MO, and *ubap2a*-i4e5 are the splice-blocking morpholinos, their knockdown efficacy was confirmed by RT-PCR

using primers nearing exon-intron regions of *ubap2a* and *ubap2b* (Supplementary Fig. S10a, g). Injections with these morpholinos resulted in aberrant transcripts that were confirmed by RT-PCR analysis. *ubap2a*-AUG MO is a translation-blocking morpholino whose knockdown efficacy had to be confirmed by either western blotting or immunostaining for the protein. The efficacy of *ubap2a*-AUG MO was confirmed by whole-mount immunostaining with Ubap2 antibody (more than 30% reduction of the Ubap2 protein (Fig. S10d-f).

Reviewer #2 (Remarks to the Author):

The authors first describe a 2 stage case-control GWAS of osteoporosis in Korean women using very small discovery (312 cases, and 983 controls) and replication (131 cases, and 1,240 control) samples. DNA samples were genotyped on Affymetrix 500K array, however, only 6,485 exonic SNPs were assessed, which seems unusually given that GWAS associations are frequently detected in non-coding regions. The authors define the significance threshold at $p < 7.7E-6$ (Bonferroni), however, none of the SNPs in the discovery study achieved significance or were suggestive (e.g. best SNP rs2781 $p = 2.3E-4$), therefore strictly speaking, there was no significant association to replicate. Subsequently, the authors combined the two cohorts and repeated the GWAS. There are several concerns about the statistical approach; a 2-stage GWAS with no significant findings followed by a combined analysis (n.b. this was not described as a meta-analysis and no covariate appears to have been included to represent the different populations combined in the analysis), therefore, the report effectively presents a very small exonic GWAS of 443 cases 2,223 controls, with one significant finding, rs2781 ($p = 6.1E-7$), without replication.

We acknowledge the reviewer's concerns. We address them as follows:

→ The GWAS of bone mineral density in the Korean research subjects (Ansung and Ansan cohort) used in this study was previously published [Cho et al., Nat Genet. 2009; 41(5):527-34. DOI: 10.1038/ng.357]. In that study, the GWAS results for the distal radius and midshaft tibia bone mineral density were reported as given below.

Table 2 SNPs showing strong evidence of association with bone density-related traits in the stage 1 (GWA) and stage 2 samples

RS ID	Sub-trait	Minor allele	Stage 1			Stage 2			Trend P value	
			MAF	Effect size ($\beta \pm$ s.e.m.)	Variation/copy	MAF	Effect size ($\beta \pm$ s.e.m.)	Variation/copy	KARE	Replication
rs7776725	RT	C	0.13	0.222 \pm 0.033	0.212	0.13	0.228 \pm 0.036	0.347	1.0E-11	1.9E-10
	TT	0.155 \pm 0.032		0.201	1.6E-06					
	CT									
rs1721400	RT	T	0.17	-0.088 \pm 0.029	-0.336	0.17	-0.110 \pm 0.032	-0.051	2.2E-03	6.0E-04
	TT	-0.149 \pm 0.028		-0.136	1.4E-07					
	CT									

Minor alleles for which the effect is estimated refer to the positive strand based on build NCBI 36. s.e.m., standard error. RT, T-score at distal radius; TT, T-score at midshaft tibia; CT, T-score at calcaneus.

→ Despite the GWAS analysis for the non-coding region having been already published, there still exists a possibility of identifying novel SNPs in the exon regions. Therefore, we conducted association studies for osteoporosis and bone mineral density using only SNPs that exist in the exon regions. As shown in Table 1, the significance level of rs2781 of the UBAP2 gene in the discovery cohort study was $p = 2.3E-4$ and the replication cohort study was $p = 9.7E-4$. The significance level in the combined analysis, $p = 5.3E-7$, could satisfy the Bonferroni significance threshold ($p < 7.7E-6$). Although our results of GWAS have limitations in terms of the small size of SNPs and in lacking replication, we conducted intensive functional studies of UBAP2 gene in cells, zebrafishes, as well as human samples, proving the possibility that UBAP2 is a susceptible gene for osteoporosis and/or bone mineral density. Through various molecular and biochemical analyses, we have demonstrated that UBAP2 is closely involved in bone remodeling. Our study lays emphasis on the possibility of the clinical usefulness of UBAP2 as a serum biomarker for diagnosis and for the evaluation of the effects of drug treatments for osteoporosis. Therefore, our findings suggest that the association study played a leading role in

identifying an important risk locus for osteoporosis that might otherwise be ignored in the conventional GWAS.

· Age was used as a covariate, however, some measure of body size should also have been included such as weight, height or body mass index; indeed both adult body size and BMI have both been shown through GWAS to be associated with SNPs in UBAP2 (rs544957562 and rs7019441 respectively).

→ According to the reviewer’s suggestion, we analyzed the association results of the rs2781 SNP by adding BMI as a covariant. The results are summarized in Supplementary Table S7.

“The significance of rs2781 for osteoporosis was $p=2.43E-6$ [OR=1.68 (CI: 2.08-4.71)] which was higher than the original results ($p = 5.3E-7$) (Table 1). In addition, upon adding BMI as a covariant, p value ($p=4.42E-7$) of rs2781 for the quantitative trait analysis of speed of sound (SOS) T-score at midshaft tibia (MT) was similar to the original results ($p = 1.1E-7$) (Supplementary Table S3).

SNP	BP	A1	TEST	NMISS	OR	L95	U95	Osteoporosis P	NMISS	BETA	SE	MT P
rs2781	33911977	C	ADD	2666	1.68	2.084	4.714	2.43E-06	3569	-0.2001	0.03955	4.42E-07

→ Of the two SNPs, only rs7019441 could be analyzed in our study cohort data. In the GWAS catalog database (<https://www.ebi.ac.uk/gwas>), the rs7019441 showed a highly significant association level ($p=5.0E-10$) with BMI as shown below.

Variant and risk allele	P-value	P-value annotation	RAF	OR	Beta	CI	Mapped gene	Reported trait	Trait(s)	Background trait(s)	Study accession	Location
rs7019441-?	3×10^{-10}		NR	-	-	-	UBAP2	Body mass index	body mass index	-	GCST009871	9:33993998
rs7019441-?	5×10^{-10}		NR	-	-	-	UBAP2	Body mass index	body mass index	-	GCST007039	9:33993998

→ In our cohort, the rs7019441 had no significant association with osteoporosis, but with the MT-SOS T-score: $p=3.92E-4$ (beta=-0.18), indicating an inverse relation of BMI with bone mineral density.

SNP	BP	A1	TEST	NMISS	OR	L95	U95	Osteoporosis P	NMISS	BETA	SE	MT P
rs7019441	33993996	A	ADD	2665	1.219	1.603	1.421	0.1554	3568	-0.18	0.05071	0.000392

· Principal component analysis (PCA) is a routinely used to correct for population stratification in GWAS and to cluster individuals by ancestry, however, this does not appear to have been done in this study. It is also not stated if any of the participants were related, how this was assessed and if necessary, how this was accounted for in the analyses.

→ We acknowledge the concern raised by the reviewer. Please note that, in a previous study [Cho et al., Nat Genet. 2009; 41(5):527-34. DOI: 10.1038/ng.357], PCA analysis of our study cohort data and comparison with other races in HapMap data were already reported. In that study, 608 patients with an identify-by-state (IBS) of 0.8 or higher were excluded to remove related participants. The published data from that study are shown below.

Supplementary Fig. 3. Multidimensional Scaling (MDS) Analysis and Principal Component Analysis (PCA). A, The MDS analysis was performed in all individuals from KARE project and 270 individuals from HapMap data using the PLINK software package (<http://pngu.mgh.harvard.edu/purcell/plink/>) and R statistics package (<http://r-project.org>). The identity-by-descent (IBD) pairwise distances among all individuals in both KARE and HapMap groups were used to construct dimensions. Individuals in KARE, Japanese (JPT), Chinese (CHB), Yoruban (YRI), and Caucasian (CEU) groups were plotted by their first and second dimension values in black, red, blue, green, and violet colored-dots, respectively. KARE samples clearly cluster together with CHB and JPT components of HapMap. B, KARE and HapMap individuals were plotted based on the first two eigenvectors obtained by PCA analysis. Individuals in KARE, Japanese (JPT), Chinese (CHB), Yoruban (YRI), and Caucasian (CEU) groups are represented by closed cyan squares, open pink squares, blue asterisks, dark green crosses, and red crosses, respectively. Again, KARE samples cluster together with CHB and JPT components of HapMap.

Supplementary Table 2. Sample size for KARE genome-wide association analysis. The final number of samples for association analyses was determined by excluding samples after each step of quality control.

	Number of samples	Remaining samples
BRLMM call rate \geq 96%	9,603	9,603
Heterozygosity \geq 30%	11	9,592
IBS \geq 0.8	608	8,984
Sex inconsistency	41	8,943
Individuals with tumor	101	8,842

· It is not clear what modality (or manufacturer) was used to classify the cases as osteoporotic – DXA, QUS, QCT? The authors next performed a GWAS of the quantitative ultrasound (QUS) phenotype

SOS. They refer to bone density; bone density measurements are based on X-ray attenuation (either QCT or DXA), whereas ultrasound parameters reflect the structural anisotropy of bone; bone properties other than density are detectable by QUS and not by DXA (e.g. elasticity and microstructure). QUS measures can be manipulated to estimate BMD (eBMD), but it is not clear that was done in this study. If the osteoporosis cases were originally defined using SOS (and not DXA), then the quantitative trait GWAS seems like duplication and it is unsurprising that it also highlighted the same genetic variant. GWAS in the literature do tend to be Caucasian-centric and a GWAS of osteoporosis in women of non-European ancestry is potentially of substantial merit and interest.

→ We acknowledge the concern of the reviewer. However, please note that the explanation for QUS measurement is given in detail in the Materials and Methods section (5.3. BMD Measurement, lines 539-552) on page 22.

· The authors describe an association with UBAP2 as novel, but an association between quantitative ultrasound measures of bone density has previously been reported with rs747091 in UBAP2 (Morris et al. Nat Genet. 2019 Feb;51(2):258-266.); rs2781 is in linkage disequilibrium $D' = -0.85$, $r^2 = 0.61$, suggesting that the finding may represent the same association signal as previously reported. The current report would be of particular value if the functional SNP had been confidently identified, however, that cannot be determined from the information available (but seems unlikely given the statistical power of the Morris et al 2019 study).

→ We thank the reviewer for raising this concern. Morris et al. (Reference 27, Morris et al., Nat Genet. 2019; 51:258-266. DOI:10.1038/s41588-018-0302-x) discovered 518 loci (301 novel) through GWAS for BMD in the 426,824 individuals, and among them 126 target genes were selected to examine the rapid-throughput skeletal phenotyping in knockout mice. In that study, UBAP2 was reported but was not selected as a study-target gene. Therefore, UBAP2 gene is not mentioned in the text and just listed as one of 301 novel loci in the supplementary data. On the other hand, our study reports the results of the functional studies of UBAP2 in vitro and in vivo models and further using human sample. Our findings explore the molecular mechanisms underlying the involvement of UBAP2 in bone remodeling, and usefulness of serum UBAP2 as a novel biomarker for the diagnosis and evaluation of the effects of drug treatments for osteoporosis.

→ Moreover, in the study by Morris et al., a highly significant association of the intronic SNP rs747091-T with heel bone mineral density was reported ($p=9E-13$). The decrease in heel bone mineral density in the rs747091-T allele is consistent with an increased risk of osteoporosis and a decrease in the MT-SOS T-score.

Variant and risk allele	P-value	P-value annotation	RAF	OR	Beta	CI	Mapped gene	Reported trait	Trait(s)	Background trait(s)	Study accession	Location
rs188795046-G	4×10^{-6}		NR	-	0.4254 SD units increase	[0.25-0.6]	UBAP2	CTACK levels	chemokine (C-C motif) ligand 27 measurement	-	GCST004420	9:34030384
rs747091-T	9×10^{-13}		0.440424	-	0.0133146 unit decrease	[0.0097-0.017]	UBAP2	Heel bone mineral density	heel bone mineral density	-	GCST006979	9:34045016

→ The association results of rs747091 in our cohort are already shown in Supplementary Table S3. The rs747091-T showed a significant association with osteoporosis [$p=1.9E-4$, $\beta=-0.20$, $OR=1.54$ (CI: 1.23~1.92) and with the MT-SOS T-score ($p=1.3E-6$). These results support the possibility that UBPA2 is a susceptible gene for osteoporosis-related traits.

· The reference that is provided to the clinical details of the GWAS cohorts (39) is not adequate and more detail is needed in the text and supplement of this paper. Also, Line 132 and table 1 state - cases, 443; controls, 2,223 were used in the GWAS, yet the abstract states the GWAS was in 3,569 women, so this cannot be reconciled.

→ We thank the reviewer for highlighting this issue. The information of clinical details of the GWAS cohorts (reference No. 39) was added in the manuscript (lines 523-531), and the number of subjects in the abstract was changed to 2,666 women (line 61).

· A series of functional analyses of the target gene (UBAP2) were performed using in vitro experiments. These appear well done and provide interesting data on the potential role of UBAP2 in bone cells. The statement is made that “UBAP2 has great potential as a novel bone biomarker for the clinical assessment of osteoporosis in women, including diagnosis and evaluation of the effects of drug treatments” – based on the evidence provided this is a substantial overstatement and needs to be revised e.g. “has great” versus “may have”.

→ We thank the reviewer for the encouragement. According to the editor’s suggestion, we changed it to “may have” (line 476).

· Furthermore, ROC curves are frequently used to show sensitivity and specificity for a clinical test. The area under the ROC curve gives an idea about the benefit of using the test; although the number of available subjects is small, this could be examined in regard to the utility of UBAP2 as a clinical biomarker for bone and in comparison to osteocalcin.

→ We thank the reviewer for the kind suggestion. ROC curve regression analysis was performed using R program of the pROC package (version 1.18) (Reference 27. Robin et al. BMC Bioinformatics 2011; 12, 77. DOI:10.1186/1471-2105-12-77) with the ELISA value data of UBAP2 and OCN in Fig. 7. The optimal cut-off point, specificity, sensitivity, and the area under the curve (AUC) were determined. The results are shown in Supplementary Fig. S15 and the detailed explanation and interpretation of the results are added in the Results (lines 321-332) and Discussion sections (lines 458- 477).

· Some of the text in the discussion is repetition of the results and more discussion is needed on the relative effect size and minor allele frequency of this variant compared to other variants for genetics of osteoporosis, pleiotropic associations seen with UBAP2 and other traits such as body size and BMI, and the biology of UBAP2.

→ According to the editor’s suggestion, we conducted the association analysis of SNPs in the UBAP2 by adding BMI as covariant. The results are summarized in Supplementary Table S7 and discussed the

results in the Discussion section (Please refer to lines 350-361).

· An important element that is missing from the report is the investigation of the different effects of the C-allele at rs2871 versus the G-allele in the functional studies.

→ We thank the reviewer for raising this concern. GTEx Portal (<https://gtexportal.org/home/>) consistently showed the differential expression between the genotypes of rs2781 SNP in various tissues and cells as shown in the figure below. The expression level of UBAP2 is higher in the C allele than G allele. These results suggest that allele difference in the rs2781 SNP may affect the expression levels of UBAP2.

Reviewer #3 (Remarks to the Author):

The manuscript "UBAP2 as a novel biomarker for the clinical assessment of osteoporosis in woman" presents convincing data that UBAP2, a protein likely involved in the ubiquitination pathway, has bone anabolic, but also pro-osteoclastogenic functions, and because of its increased concentration in peripheral blood samples might be a useful biomarker to detect osteoporosis. The authors show that knockdown of UBAP2 results in bone mineralization defects in MC3T3-E1 osteoblast cell culture, and that osteoclastogenesis is enhanced after knockdown in primary-cultured monocytes. They investigate the function of *ubap2* in zebrafish by using a morpholino knockdown approach and combine their in vitro and in vivo results with the investigation of UBAP2 levels in bone marrow and peripheral blood samples of patients. To my knowledge, a role in bone homeostasis and bone pathology has not been demonstrated for UBAP2 before, thus the results presented are novel and definitely interesting.

However, some questions remain:

1) The function of *ubap2* in formation and calcification of bone and cartilage in zebrafish remains somewhat enigmatic. Alcian Blue staining and Alizarin Red staining suggest reduced formation. However, certain improvements should be made to confirm this finding. Because of a genome duplication event in teleost fish species, many genes have two paralogues and according to ZFIN (zebrafish information network), this also seems to be true for *ubap2*. Which gene (*ubap2a* or *ubap2b*) are we dealing with here? Testing of both paralogs may be necessary or an explanation should be provided. Also, morpholino knockdown approaches have their caveats, and have to make use of stringent controls, to exclude developmental delay phenotypes. Importantly, rather than uninjected controls, control-morpholino-injected embryos should be used (e.g. mismatch morpholino, see Stainier et al. 2017 PlosGenetics). In addition, a second morpholino (e.g. translation blocking morpholino) or alternatively a mutant is required to validate the data, as morpholino usage often produces off-target effects. While I do not doubt the results presented with the splice blocking morpholino, I suggest to add a second morpholino and to control for unspecific effects by using a mismatch morpholino as a control. If the same effects are seen, the phenotype should be rescued by injection of *ubap2a/b* mRNA. Alternatively, mutant zebrafish, created by the Sanger institute, or by injection of *ubap2a/b* targeting sgRNAs and Cas9 should be analyzed.

→ We fully agree with the reviewer's concerns on off-target effects of morpholinos. To address this issue in the revised manuscript, we carried out additional experiments according to the reviewer's suggestions as follows:

1) Eye size measurement (Fig. 3): while the lengths of the Alizarin-positive notochord were significantly reduced in the *ubap2a* and *ubap2b*-morpholino RNA injected conditions compared to control (Fig. 3a, b, c), the eye sizes of the injected larvae remained similar among experimental groups (Fig. 3a, b, d), suggesting that shortening of the notochord upon morpholino injection is not due to a general developmental delay associated with non-specific toxicity of morpholinos.

2) Control morpholino and second morpholinos for each gene (Fig. 3, Supplementary Figs. S10~S12): instead of using 'uninjected controls', we used the standard control morpholino RNA from Gene Tools that is widely used in zebrafish morpholino studies as a negative control (<https://www.genetools.com/content/negative-control-morpholino-oligos>).

In addition, we designed a second translation-blocking morpholino against *ubap2a* (Supplementary

Figs. S10a, b) and validated its efficacy using the UBAP2 antibody by whole-mount immunostaining (Supplementary Figs. S10d-f). For *ubap2b* knockdown, we designed two independent splice-blocking morpholinos and validated their efficacy using RT-PCR and direct sequencing of aberrant PCR bands (Supplementary Figs. S10g-i). Two independently designed morpholinos for each gene showed identical bone formation defects in the pharyngeal skeleton (Fig. 3 and Supplementary Fig. S11). Furthermore, knockdown of *ubap2a* or *ubap2b* displayed similar bone formation defects, suggesting that both genes play an important role in the bone formation.

3) Rescue experiments (Supplementary Fig. S11): the gene specific effects of *ubap2a* or *ubap2b* knockdown were demonstrated through the rescue experiments with the addition of mRNAs. The abnormal bone phenotypes and notochord length in the *ubap2a* and *ubap2b* MO-injected larvae were completely rescued by co-injection of mRNAs of *ubap2a* or *ubap2b*, respectively (Fig. 3). Taken together, our results indicated that the bone formation defects caused by knockdown of *ubap2a* or *ubap2b* are specific and are not the unwanted off-target side effects of morpholinos.

2) The authors discuss a potential function of Anxa2 downstream of UBAP2. In order to provide mechanistic insight into UBAP2's action I suggest to investigate the hypothesis of Anxa2 in cell culture assays. For example, Anxa2 protein and/or transcript levels could be evaluated after UBAP2 knockdown, and reduced matrix formation as well as increased osteoclast formation could be rescued with overexpression of Anxa2.

→ We thank the reviewer for the kind suggestion. However, when we investigated whether UBAP2 expression could influence Anxa2 expression, unexpectedly, we could not find any correlation between these two molecules. To elucidate the possible molecular mechanisms of UBAP2 in the regulation of bone remodeling, we performed *in silico* pathway network analysis and found the candidate molecules that are associated with UBAP2 in osteoclastogenesis: E-cadherin (*CDH1*) and Fra-1(*FOSL1*). Manipulation of expression of these three genes, *UbAP2*, *Cdh1*, and *Fosl1* revealed that *Ubp2* knockdown caused osteoclastogenesis augmentation while *Cdh1* or *Fosl1* knockdown caused osteoclastogenesis reduction. Double knockdown of *Ubp2* and *Cdh1* or *Ubp2* and *Fosl1* resulted in a significant reduction of osteoclastogenesis. Taken together, our results suggest that UBAP2 might play a pivotal role in osteoclastogenesis through the regulation of E-cadherin and Fra-1. The results are shown in Fig.5 and described and discussed in detail in the Results (lines 267-300) and Discussion sections (lines 402-438).

3) line 140 onwards. The sentence is a little hard to understand, maybe the content can be split to two sentences?

→ According to the editor's suggestion, we have revised the sentence and shortened it.

4) line 214 onwards. I am afraid I do not fully understand the meaning of this sentence (calcific deposition per se). Can the authors please rephrase the sentence?

→ According to the editor's suggestion, the sentence is deleted.

5) please use bglap or ocn throughout the text (mentioning the alternative name once when the gene is mentioned first).

→ *Bglap* is gene name of osteocalcin. We address it on page 8, line 179: “*Bglap* encoding Osteocalcin (OCN)”

6) Please clarify for the readers whether OCN levels are up or down (in previous studies of osteoporotic women).

→ To address the reviewer’s concern, we added the following sentence with the reference 26 on page 13: “Serum OCN is upregulated in patients with osteoporosis and has long been used as a diagnostic biomarker for primary osteoporosis in women.²⁶”

7) line 265. Numerous genetic risk factors are responsible... (add the word 'are')

→ As per the reviewer’s suggestion, we have added the word “are”.

8) line 282 and 283. "We also..." used twice, maybe replace?

→ As per the reviewer’s suggestion, we have revised it.

9) line 286. hematopoietic, instead of hemopoietic

→ As per the reviewer’s suggestion, we have rectified it to hematopoietic.

10) please use the term bone mineral density instead of bone density

→ According to the editor’s suggestion, we have changed “bone density” to “bone mineral density” in the entire text.

11) line 497. Were the primers used for qRT-PCR exon spanning? Was efficiency testing performed?

→ Please note that the primer sequences of each gene were designed to contain at least two exons and the efficiency was tested by agarose gel-loading.

12) line 606. The experiments were repeated independently at least three times... Are the results shown representative results of one experiment or the average of three experiments?

- All qRT-PCR results are represented as the average of three experiments and the staining and immunoblotting results are shown by one representative result.

13) The figure legends contain a lot of material and methods information (e.g. the cells were incubated with 10% cetylpyridium chloride at room temperature). If this is not a journal requirement, please move this sort of information to the material & methods section.

→ According to the editor’s suggestion, the figure legends of some figures (Fig. 1, Fig. 2, and Supplementary Figures) have been shortened.

14) line 799. All experiments were repeated three times with duplicated analyses. - What does duplicated analyses exactly mean?

→ To address the reviewer's concern, the "duplicated analyses" means that the samples for analysis were duplicated in each experiment. The words are changed to "duplicated samples".

15) Fig. 5 and Fig. 6 - aren't controls also patients? maybe the legend could compare healthy versus osteoporotic or alike?

→ Please note that in Figs. 5 and 6, our clinical data are based on the comparison between healthy people with normal BMD and patients with osteoporosis.

16) Reporting summary: isn't the human data used clinical data?

→ We have filled in the part of 'Human research participants' of the reporting summary instead of the part of 'Clinical data, because this study is not clinical study.

17) It seems there are no FACS gating plots - does this section need to be answered (e.g. on contour lines)? If so, the plots would need to be shown.

→ To address the reviewer's concerns, the results of FACS gating plots are added in Supplementary Fig. S5.

REVIEWER COMMENTS

Reviewer #1 (Remarks to the Author):

The authors showed that Western blotting analysis of Ubap2 in the cell-culture media of MC3T3- 126 E1 cells after osteoblast differentiation induction for 4 days, where a cytoplasmic protein beta actin is also present, which could indicate this protein in cytosolic, not secreted.

There is no evidence that Ubap2 is secreted and it is not sure how it got into circulation and detected?

The possibility of a false positive or non specific or cross reactive antibody of Ubap2 by ELISA exists, in which the rest of this paper depends on this assumption.

Reviewer #3 (Remarks to the Author):

The authors addressed my concerns adequately. They checked for the expression of the two paralogues of UBAP2 in zebrafish, and also knocked down both genes with the help of two different morpholinos. In addition, the effects of the morpholino treatment were rescued.

I think some controls are missing (figure wise), and I am sure the authors will be able to add the respective data:

1) Fig. S9 a-d: Please show sense controls (of the probes used) for the RNA in situ hybridizations for at least one stage (e.g. 24 hpf). This is to exclude that the staining is unspecific, and a sense control is a very convincing control.

2) Fig. S9e,f: Please, in addition to the "without secondary antibody" control, show a "without primary antibody" control, in order to test for the unspecific fluorescence that is produced due to the secondary antibody in the tissue. This is the more stringent and important control and it is also necessary to judge about the results shown in Fig. S10e (Knockdown via Translation blocker), since no Western Blot is shown.

3) Fig. S10i: Please indicate the DNA ladder fragment sizes on the left.

4) The authors talk about a 'shorter notochord' after knockdown, but what they mean is a shorter mineralized notochord (the notochord is of course much longer than what is stained by alizarin red at 6 dpf). It should be called (in graphs and in the text): calcified notochord length, or length of alizarin red stained notochord.

5) Efficiency testing of primers does require quantitative assessment of fragment amplification in qPCR. Primer efficiency is recommended to be in 90 – 110 %. The authors stated that efficiency was tested based on gel electrophoresis; this can however only test for the correct fragment size.

Please mention in the Materials & Methods that the used qRT-PCR primers were covering two exons and that they were not tested for their efficiency (alternatively do the test), but for correct product size.

6) Material and methods: Which primer sequences were used to clone the two genes full cDNA into the expression vector (for rescue experiments)? Please add.

7) As a general comment to the morpholino knockdown experiments: combining both ubap2a and ubap2b knockdown might have resulted in much stronger phenotypes and should be considered (no requirement from my side).

Response to Reviewers' comments

We thank the reviewers for their valuable comments on our manuscript. We have revised the manuscript according to their comments and provided detailed, point-by-point responses below. All changes in the text appear in red. The manuscript has been edited and proofread by native-English-speaking, professional scientists at EDITAGE, a scientific research paper editing company.

Reviewer #1:

The authors showed that Western blotting analysis of Ubap2 in the cell-culture media of MC3T3-E1 cells after osteoblast differentiation induction for 4 days, where a cytoplasmic protein beta actin is also present, which could indicate this protein in cytosolic, not secreted.

Response: We thank the reviewer for pointing this out. We apologize that the results might have been unclear in Supplementary Fig. S7b (the previous version). The upper panel was the western blotting result of Ubap2 in the cell-culture media of MC3T3-E1 cells, and the lower panel was the western blotting result of β -actin in whole cell lysates of MC3T3-E1 cells (for the loading control). These results indicate that despite no evidence that Ubap2 is secreted, Ubap2 might have also been present extracellularly. We deleted Fig. S7b to avoid a misunderstanding of the results.

To verify the possibility of a false positive or non-specific or cross-reactive antibody of Ubap2, we purchased two other Ubap2 antibodies that recognize different epitope regions of Ubap2 (Abcam; ab197083 and Bethyl laboratories. Inc.; A304-626A) with different epitopes from the existing antibody (Abcepta; AP12773a), and their specificities were tested using western blot analysis in MC3T3-E1 cells. The specificities of all Ubap2 antibodies were confirmed (Supplementary Fig. S9). The results were same and consistent in three Ubap2 antibodies: Ubap2 level in the cell lysates of MC3T3-E1 cells was increased by the induction of osteoblast differentiation.

Fig. S9. Specificity test for the Ubp2 antibodies in pre-osteoblast MC3T3-E1 cells. For osteoblast differentiation induction, cells were incubated with the osteoblastic medium containing ascorbic acid and β -glycerophosphate for 4 days. The three kinds of Ubp2 antibodies (Abcepta, Abcam, and Bethyl Laboratories, Inc.) that recognize different epitope regions of Ubp2 protein. Osteocalcin (OCN) and β -actin antibodies were used as controls.

There is no evidence that Ubp2 is secreted, and it is not sure how it got into circulation and detected?

Response: We acknowledge the reviewer's concern that there is no evidence that Ubp2 is secreted. Ubp2 predominantly exists in the cytosol (in Fig. S7). Our western blotting results below show that Ubp2 is detected in the cell-culture media and whole cell lysates of pre-osteoblast MC3T3-E1 cells. Ubp2 protein level in cell-culture media is increased by osteoblastogenic induction (Mock), but knockdown of *Ubp2* by RNA interference (*shUbp2*) resulted in a decrease of Ubp2 (Please refer to the supporting figure below). These results suggest that Ubp2 may exist in cell-culture media, which might explain how Ubp2 is detected in blood plasma by ELISA (Fig. 7 and S16).

Supporting Figure. Pre-osteoblast MC3T3-E1 cells were cultured in normal media (Control) and osteoblastic media (Mock, Induction,) and cells were transfected with a control vector (shVec) and short hairpin RNA (*shUbp2*) in the osteoblastic media. The supernatant proteins were concentrated using the trichloroacetic acid precipitation method. Western blotting of anti-Ubp2 antibody was performed in whole cell lysates and cell culture media. Osteocalcin (OCN) and β -actin antibodies were used as controls. Ponceau S staining was used as the loading control.

OCN is one of the key players in bone endocrinology expressed and secreted by osteoblasts during the bone formation phase of bone remodeling (Hauschka PV, et al., *Physiol Rev.* 1989; 69:990–1047., Hamdi RA, et al., *J Fac Med Baghdad.* 2013; 55(2):166-69.). Alkaline phosphatase (ALP) is also present in the bloodstream and on the external surface of most cells. The total ALP test can be used to diagnose bone problems such as osteoporosis (Fink HA, et al., *Osteoporos Int.* 2016; 27:331–8., Kuo, et al., *Biomarker Research.* 2017; 5:18). Procollagen type 1 C-terminal Propeptide (PICP), a single protein with 115 kDa in serum, is used as a biomarker of bone formation (Eriksen EF., et al. *J Bone Miner Res.* 1993; 8:127–32.).

It is unknown how ALP and PICP can be detected in serum. Likewise, it is unclear how UBAP2 is present extracellularly and detected in blood plasma. Based on the ELISA results in blood plasma, UBAP2 can be used as a biomarker of osteoporosis.

The possibility of a false positive or non specific or cross reactive antibody of Uba2 by ELISA exists, in which the rest of this paper depends on this assumption.

Response: We thank the reviewer for raising this concern. To test whether the UBAP2 ELISA results in Fig. 7 might be caused by a false positive or non-specific or cross-reactive antibody reaction, we conducted the same ELISA experiment using other ELISA kit for UBAP2 (antibodies-online GmbH; ABIN6971034; Aachen, Germany) in the available same human plasma samples. The level of UBAP2 was significantly higher in the patient samples compared to the normal control samples. The results were consistent with those of the previous UBAP2 ELISA kit (USCN Life Science Technology; Wuhan, China) (Fig. 7 and Fig. S16).

The standard curve data and ELISA results of two different UBAP2 antibody-used ELISA kits are shown below.

(1) Standard curve data and ELISA result in Fig. 7.

(2) Standard curve data and ELISA result in Fig. S16.

The above results indicate that the UBAP2 antibody in the ELISA specifically detected the plasma UBAP2. In addition, as shown in Fig. S17, the area under the curve (AUC) values representing overall accuracy for the diagnostic tests using UBAP2 and OCN were determined as 0.727 and 0.702, respectively, indicating that the overall accuracy for osteoporosis diagnosis is comparable between the two.

Reviewer #3:

The authors addressed my concerns adequately. They checked for the expression of the two paralogues of UBAP2 in zebrafish, and also knocked down both genes with the help of two different morpholinos. In addition, the effects of the morpholino treatment were rescued.

I think some controls are missing (figure wise), and I am sure the authors will be able to add the respective data:

1) Fig. S9 a-d: Please show sense controls (of the probes used) for the RNA in situ hybridizations for at least one stage (e.g. 24 hpf). This is to exclude that the staining is unspecific, and a sense control is a very convincing control.

Response: We thank the reviewer for pointing this out. We conducted *in situ* hybridizations with sense control probes. Representative images of the RNA *in situ* hybridization of *ubap2a* and *ubap2b* at 2 dpf were added at the bottom of Figs. 10a and 10b (Fig. S9 is changed to Fig. S10). As expected, the hybridization of sense probes for *ubap2a/b* did not detect any RNA expression, indicating specific staining of the anti-sense RNA probes for *ubap2a/b*.

2) Fig. S9e,f: Please, in addition to the "without secondary antibody" control, show a "without primary antibody" control, in order to test for the unspecific fluorescence that is produced due to the secondary antibody in the tissue. This is the more stringent and important control and it is also necessary to judge about the results shown in Fig. S10e (Knockdown via Translation blocker), since no Western Blot is shown.

Response: We conducted a control experiment of immunostaining "without primary antibody" according to the reviewer's recommendation. In Figs. S10e and 10f 10b (Fig. S9 is changed to Fig. S10), the absence of a primary antibody did not detect any fluorescence staining, except weak auto-fluorescence in the lens and the yolk, indicating the specific detection of Ubp2.

3) Fig. S10i: Please indicate the DNA ladder fragment sizes on the left.

Response: The DNA ladder sizes were presented on the left of the gels in Fig. S11i (Fig. S10 is changed to Fig. S11).

4) The authors talk about a 'shorter notochord' after knockdown, but what they mean is a shorter mineralized notochord (the notochord is of course much longer than what is stained by alizarin red at 6 dpf). It should be called (in graphs and in the text): calcified notochord length, or length of alizarin red stained notochord.

Response: According to the reviewer's recommendation, we corrected the terminology from "length of notochord" to "length of ARS-stained notochord" in the main text, graphs, and legends of Fig. 3 and Fig. S12.

5) Efficiency testing of primers does require quantitative assessment of fragment amplification in qPCR. Primer efficiency is recommended to be in 90 – 110 %. The authors stated that efficiency was tested based on gel electrophoresis; this can however only test for the correct fragment size.

Please mention in the Materials & Methods that the used qRT-PCR primers were covering two exons and that they were not tested for their efficiency (alternatively do the test), but for correct product size.

Response: According to the reviewer's recommendation, we have added a sentence in the Methods & Materials section 5.15 as follows:

"These semi-qRT-PCR primers used in Fig. S11g-i covered two exons for checking the correct PCR fragment sizes; they were not tested for their efficiency." (Page 29, Lines 730 to 731)

6) Material and methods: Which primer sequences were used to clone the two genes full cDNA into the expression vector (for rescue experiments)? Please add.

Response: We replaced the information of the cDNA clone and primer sequences in the Methods & Materials section as follows:

"A full-length cDNA clone for zebrafish *ubap2a* was purchased from Open Biosystems (Clone ID: 6790255; Huntsville, AL, USA). *ubap2b* cDNA was amplified using a primer pair (forward primer 5'-CGG AGA GGA CTT TTC TAT TTT G-3'; reverse primer 5'-CGT GTT AGA GGC AGG ACA GAG CG-3') with a PCR cycle [94°C, 5 min, 30 cycles of (94°C, 30 sec; 55°C, 30 sec; 72°C, 3 min), 72°C, 7 min], followed by subcloning into the pCRII TOPO vector (Thermo Fisher Scientific Inc.). Full-length cDNAs of *ubap2a* and *ubap2b* were subcloned into a pCS2+ vector, and mRNAs were synthesized using the mMACHINE™ SP6 Transcription Kit (Invitrogen).

7) As a general comment to the morpholino knockdown experiments: combining both *ubap2a* and *ubap2b* knockdown might have resulted in much stronger phenotypes and should be considered (no requirement from my side).

Response: We appreciate the reviewer for this comment. Double knockdown of *ubap2a* and *ubap2b* may lead to much stronger bone phenotypes. However, it might cause unusual severe developmental delay or false positive phenotypes due to morpholino toxicity. Either a single knockdown of *ubap2a* or *ubap2b* showed obvious abnormal phenotypes on bone formation.

REVIEWER COMMENTS

Reviewer #1 (Remarks to the Author):

The authors have addressed questions and the paper has been improved.

Response to the Reviewers' comments and suggestions

Reviewer #1 (Remarks to the Author):

The authors have addressed questions and the paper has been improved.

We thank the Reviewers for their valuable comments on our manuscript.